



**Diversity and assembly processes of microeukaryotic**
**community in Fildes Peninsula Lakes (West Antarctica)**
Chunmei Zhang[a,b], Huirong Li[a,c,d], Yinxin Zeng[a,c,d], Haitao Ding[a,c,d], Bin Wang[e],
Yangjie Li[e], Zhongqiang Ji[e], Yonghong Bi[b*], Wei Luo[a,c,d*]
[a] Key Laboratory for Polar Science, Polar Research Institute of China, Ministry of
Natural Resources, Shanghai 200136, China
[b] State Key Laboratory of Freshwater Ecology and Biotechnology, Institute of
Hydrobiology, Chinese Academy of Sciences, Wuhan 430072, China
[c] Antarctic Great Wall Ecology National Observation and Research Station, Polar
Research Institute of China, Ministry of Natural Resources, Shanghai 200136, China
[d] School of Oceanography, Shanghai Jiao Tong University, Shanghai 200030, China
[e] Key Laboratory of Marine Ecosystem Dynamics, Second Institute of Oceanography,
Ministry of Natural Resources, Hangzhou 310012, China
**\*Corresponding Author:** biyh@ihb.ac.cn, luowei@pric.org.cn



## Abstract

The diversity, co-occurrence patterns and assembly processes of microeukaryotes in Antarctic freshwater lakes are not well understood, despite its wide distribution and ecological importance. This study used Illumina high-throughput sequencing to explore five freshwater lakes' microeukaryotic communities on the Fildes Peninsula during three summer seasons. A total of 33 phyla were detected, with the phytoplankton occupying the highest percentage of sequences (accounting for up to 98%). Meanwhile, the main dominant taxa were Chrysophyta, Chlorophyta, and Cryptophyta. Alpha diversity varied among lakes, with Changhu (CH), Kitec (KT) lake having higher values, and Yue Ya (YY) lake having the lowest value. There were significant differences in microeukaryotic communities between lakes, with spatial and temporal variation in the relative abundance of dominant taxa ($P<0.05$). Environmental variables only explained about 30% of the variation in community structure. In the co-occurrence network, microeukaryotes tended to be more symbiotic than competitive with each other (positive correlation 82% vs. negative correlation 18%), with only 8% of OTUs significantly associated with environmental factors. The neutral community model found that neutral processes explained more than 56% of the community variation. The stochastic processes (e.g., homogenizing dispersal and undominated process) predominated in community assembly than the deterministic processes. These findings revealed the diversity of the microeukaryotic community and have important implications for understanding the community assembly in the freshwater lakes of the Fildes Peninsula (Antarctica).

**Keywords:** Microeukaryotic community; Diversity; Co-occurrence network; Stochastic processes; Fildes Peninsula lakes.



## 1 Introduction

The Fildes Peninsula locates in the southwestern part of King George Island of the West Antarctic region, which has a high concentration of Antarctic scientific research stations worldwide and is commonly free of ice during summer. It is the largest ice-free area (40 km$^2$) on King George Island, South Shetlands. This area falls within the "maritime Antarctic" with precipitation of 400-600 mm year$^{-1}$ and an average temperature of -3 ℃ (Holdgate 1977). Nevertheless, permafrost and periglacial processes occur (Barsch and Caine 1984). The waters are affected by natural factors such as the sea, animals, and anthropogenic pressures in solid, volatile, and fluid wastes (Kawecka et al., 1998). Lakes in Fildes Peninsula, along with those found in other ice-free areas in Antarctica, represent the year-round liquid water reservoirs on the continent (Lyons W B  et al., 2007; Priscu 2010). Antarctica lake systems are sentinels for climate change and contain chemical elements and microorganisms of global relevance (Marsh et al., 2020; Wilkins et al., 2013). Because of their physical stability, such lakes have been identified as model systems for inferring biogeochemical processes within water columns (Comeau et al., 2012). Most Antarctic lakes are ultra-oligotrophic to oligotrophic, which only allow a few species to adapt to such extreme environments resulting in truncated simplified food webs (Izaguirre et al., 2020).

As an essential component of microorganisms in Antarctic freshwater lake systems, microeukaryotes has shown critical roles in the ecosystem services, acting as the main food source and the primary contributors to material circulation (Mo et al., 2018b; Moreno-Pino et al., 2016; Zeng et al., 2014). The small size, short generation time, rapid growth, sensitivity to environmental conditions, and genetic plasticity render them capable of quick reflection to environmental changes (Karimi et al., 2017). Hence, they are excellent bioindicators of the impact of environmental perturbations and ecosystems quality (Bouchez et al., 2016). The Ciliophora, Cryptomycota, Chlorophyta, and Bacillariophyta have been detected in the Freshwater Glacier Lake, East Antarctica, the biodiversity seem to be affected by the





temperature and salinity (Lopez-Garcia et al., 2001). A pretty low number of taxa, the
abundance of diatom genera such as *Nitzschia*, *Achnanthes*, and *Navicula*, etc., have
been investigated from the periodically brackish water ponds near the Polish Antarctic
Station on King George Island (Kawecka et al., 1998). The microeukaryotic
communities of shallow lakes from the Antarctic Peninsula are influenced by nutrient
and surrounding inputs (Mataloni et al., 2000). However, the spatio-temporal
variation, co-occurrence pattern, and community assembly of microeukaryotes in
Antarctic (Fildes Peninsula) freshwater lakes have been rarely reported.

Deterministic and stochastic processes have been considered the two main

ecological processes in community assembly (Ofiteru et al., 2010). Deterministic
processes are based on ecological niche theory; some deterministic factors
(environmental conditions and species interactions) influence and determine
community assembly (Powell et al., 2015). Stochastic processes are based on the
neutral theory, which believes that random birth or death, drift, and dispersal events
also play an essential role in community composition (Bahram et al., 2016).
Deterministic processes (selection) are prevalent in building whole ecosystem
communities (Liu et al., 2020a), selection leads to species classification, and applying
similar habitats results in similar community assemblages. Although other studies
support a role for stochastic processes (drift and dispersal) in community assembly,
dispersal is the movement of species in spatial location, and drift is associated with
the relative abundance of species (Massana and Logares 2013; Wu et al., 2019).
Stochastic processes account for up to 95% of the microeukaryotic community
assembly mechanism in a set of lakes in Eastern Antarctica (Logares et al., 2018).
Nonetheless, the study of microeukaryotic diversity and its community assembly
processes in Antarctica still require further investigations for a more comprehensive
view.

Few studies have been conducted on microeukaryotic diversity and community

assembly processes of the freshwater lakes in the Fildes Peninsula, Antarctica. Our
study attempted to analyze the microeukaryotic samples of five freshwater lakes from



three summers using high-throughput 18S rRNA sequencing. We aim to (I)
understand the diversity and co-occurrence of microeukaryotes; and (II) to explore the
influencing factors and their community assembly processes.

**2 Material and Method**

2.1 Sampling collecting
Field samples were collected on the 34th (2017/2018), 35th (2018/2019), and
36th (2018/2019) Chinese Antarctic Research Expedition (CHINARE) in January
2018 (34th), December 2018 (35th), and December 2019 (36th), respectively. The
investigations were conducted in the Chinese Great Wall Station area at King George
Island, the largest island in southern Shetland Island. Surface water samples were
collected from five lakes, Changhu (CH), Kitec (KT), Xihu (XH), Yanou (YO), and
YueYa (YY) (Fig. 1). Moreover, the physio-chemical parameters were measured
synchronously.
Lake Chang Hu (CH) is a narrow strip shape, surrounded by bulges, with major
inputs from surrounding glacial melting water. Lake Kitezh (KT) is the closest to the
Corinthian ice cap and is the source of drinking water for the Chilean station, near the
airport for access to the Fildes Peninsula in Antarctica. The KT is the largest lake in
this investigation area. Lake Xi Hu (XH) is the drinking water source area for the
Great Wall Station scientific expedition station. Lake Yann Ou (YO) is surrounded by
mountains and snow-covered, with moss and lichen growing in the soil. It is the
smallest lake of this investigation area and is relatively sensitive to the effects of
scientific expeditions. Lake Yue Ya (YY), situated on Ardley Island, is far from
human activities but influenced by penguins dwelling on the island, which brings
massive penguins excrements inputs.
Water temperature (WT), pH, and salinity (Sal) were measured using a YSI
Model 30 (Yellow Springs Instruments, Yellow Springs, USA). Chlorophyll a (Chl a)
was extracted with acetone and measured spectrophotometrically. Nutrient, including
ammonia ($NH_4^+$), Nitrite ($NO_2^-$), silicate ($SiO_3^{2-}$) and phosphate ($PO_4^{3-}$) were





measured spectrophotometrically with a continuous flow autoanalyzer Scan++ (Skalar,
the Netherlands) after filtering water through 0.45 µm cellulose acetate membrane
filters (Whatman) as described by (HP Hansen and F Koroleff 1999).

## 130 2.2 PCR and Illumina MiSeq

For Illumina MiSeq2000, 1 L surface seawater was collected and prefiltered
through a 20-µm mesh sieve to remove most of the mesozooplankton and large
particles, then directly filtered through a 0.2 µm pore size nucleopore membrane filter
(Whatman). The filters were frozen at –80°C in CTAB buffer until laboratory
experiments. DNA extraction was performed as described by (Luo et al., 2015).
PCR was performed using primers with barcode flanking the hypervariable V4
region of the 18S rRNA gene: 3NDf (Charvet et al., 2012) with the reverse primer
V4_euk_R2 (Brate et al., 2010). Polymerase chain reactions (PCRs) were conducted
in 20 µL reactions with 0.2 µM each primer, 10 ng of template DNA, 1 × PCR buffer,
and 2.5 U of Pfu DNA Polymerase (Promega, USA). The amplification program
consisted of an initial denaturation step at 95 °C for 2 mins, followed by 30 cycles of
95 °C for 30 s, 55°C for 30 s, and 72 °C for 30 s, and a final extension of 72 °C for 5
min. PCR products were pooled and purified using the DNA gel extraction kit
(Axygen, Hangzhou, China). The DNA concentration of each PCR product was
determined using a Quant-iT PicoGreen double-stranded DNA assay (Invitrogen,
Germany) and was quality controlled on a TBS-380 Mini-Fluorometer (Turner
Biosystems, Sunnyvale, CA, USA). Finally, amplicons of all samples were pooled in
equimolar concentrations.
We used QIIME default parameters for quality filtering (reads truncated at first
low-quality base and excluded if: (1) overlap ≤10bp while the coupled reads were
assembled into one single sequence, (2) less than 80% of reading length was
consecutive high-quality base calls, (3) more than 1 errors were present in the bar
code, (4) the length was less than 50 bases (Caporaso et al., 2010). We picked
operational taxonomic units (OTUs) with a 97% similarity cut-off using available



reference UPARSE version7.1 (http://drive5.com/uparse/). Reads that did not match
any sequences in the reference database at ≥ 97% identity were clustered de novo.
The taxonomic identity of eukaryotic representative sequences was performed using
RDP classifier against the SILVA database (version 132 NR) (Quast C et al., 2013) at
a bootstrap cutoff of 80%.

## 2.3 Community composition and diversity

The OTUs and Shannon index (H) were measured using the "vegan" R package
based on the OTUs table, respectively. The nearest-taxon index (NTI) was used to
measure the degree of phylogenetic clustering of taxa on a within-community scale
for communities. High or positive values indicated clustering taxa across the overall
phylogeny, while lower negative values indicated overdispersion of taxa across the
phylogeny (Horner-Devine and Bohannan 2006). The nearest taxon index (NTI)
quantifies the number of standard deviations that the observed MNTD is from the
mean of the null distribution with 999 randomizations in the "Picante" R package.
Non-metric multidimensional scaling (NMDS) of microeukaryotic communities
was performed with the relative abundance of OTUs (Roberts 2013). Analysis of
similarity (ANOSIM) investigated differences in the microeukaryotic communities
between groups. The unweighted pair-group method with arithmetic means (UPGMA)
was used to determine the similarity between samples by clustering analysis
according to community composition similarity. These analyses were performed in the
R package "Vegan" and "Phangorn". All calculations were based on similarity
matrices calculated with the Bray-Curtis similarity index.

## 2.4 Influencing factors of the community structure

Canoco 4.5 software (Braak and Smilauer 2002) was used to rank species and
environmental factor data, and the ranking model was determined by de-trending
correspondence analysis (DCA) of OTUs data. All environmental factors, except pH,
were  log  (x+1)  transformed  before  analysis  to  improve  normality  and



homoscedasticity. To reduce multicollinearity among environmental factors, all
variance inflation factors (VIFs) were kept below 10. The environmental factors
influencing the composition of the microeukaryotic community were selected by 999
Monte Carlo permutation tests at the significant level ($P<0.05$). In addition, the
relative importance of water temperature, physicochemical factors, and nutrients was
assessed using the variation partitioning analysis (VPA).

## 2.5 Co-occurrence Network Analysis

The samples collected were performed by co-occurrence network analysis. To
reduce the complexity of the data sets, OTUs represented Occurred in at least 5
samples were retained to construct networks. Only robust ($|r| > 0.6$) and statistically
significant ($P<0.05$) correlations were incorporated into network analyses. Finally,
network visualization was conducted using Gephi software (Bastian M et al., 2009).
Previous studies identified potential keystone taxa as nodes with degree $> 30$ and
betweenness centrality $< 5000$ (Ma et al., 2020; Zhang et al., 2021a).

## 2.6 Ecological community assembly analysis

The Neutral community model (NCM) can measure the potential role of
stochastic processes in the assembly of microeukaryotic communities based on the
relationship between OTUs frequency and relative abundance (Chen et al., 2019). The
model is derived from neutral theory (Zhou et al., 2014). The parameter Nm
represents the metacommunity size, and $R^2$ represents the degree of fit to a neutral
model.
To further evaluate the contributions of deterministic and stochastic processes to
community assembly, the Stegen null model was used (Stegen et al., 2012). The β-
nearest taxon index (βNTI) was calculated using phylogenetic distance and OTUs
abundance (Stegen et al., 2013; Webb et al., 2002). The relative contribution of
variable selection and homogeneous selection was estimated from the percentage of
pairwise comparisons whose βNTI were $> 2$ and $< -2$, respectively. We further





calculated the Bray-Curtis-based Raup-Crick index ($RC_{bray}$) to investigate pairwise
comparisons that deviated from selection (Evans et al., 2017; Stegen et al., 2013).
Integrated with the value of $|RC_{bray}|$, the underlying community assembly processes
could be homogenizing dispersal ($|\beta NTI| < 2$ and $RC_{bray} < -0.95$), dispersal limitation
($|\beta NTI| < 2$ and $RC_{bray} > +0.95$) and undominated processes (i.e. weak selection,
weak dispersal, diversification, and drift processes) with $|\beta NTI| < 2$ and $|RC_{bray}| <$
0.95. The null community of all the samples was randomized 999 times to obtain
average null expectations.
**3 Result**
3.1 Physico-chemical properties

The water temperatures (WT) of all five lakes had similar values as 0.90°C to

7.14°C (Table S1), while the YO lake was significantly higher than other lakes
($P<0.05$). Nutrient values were low with nitrite ($NO_2$-N), ammonium nitrogen ($NH_4^+$),
and phosphate ($PO_4$-P) concentrations with 0.00~0.15 µM L$^{-1}$, 0.05~0.74 µM L$^{-1}$, and
0.02~2.29 µM L$^{-1}$, respectively. YY lake had higher concentrations of phosphate,
ammonium nitrogen, and nitrite, while XH had lower nitrite and phosphate. Silicate
($SiO_3^{2-}$) varied from 1.43 to 51.5 µM L$^{-1}$, with the highest value in CH and lowest
value in YY. The range of Chl a was 0.25~2.11 µg L$^{-1}$, with the YY highest and the
CH lowest. pH ranged from 7.65 to 8.27. Salinity was 0.00-0.14, which in YO lake
exhibited significantly lower ($P<0.05$).
3.2 Diversity and composition of microeukaryotic community

A total of 726,700 valid sequences of the 18S rRNA gene in all samples was

obtained, and the average length of the acquired reads was 443 base pairs. These
sequences clustered into 547 OTUs at 97% similarity level, distributed among 33
phyla. The Good's coverage values were above 99.9%, confirming that the libraries
could represent most species in these lakes.

A total of 10 dominant phyla were identified, accounting for 96.02% sequences





in CH, 97.01% in KT, 98.30% in XH, 94.19% YO, and 98.27% YY. These dominant
phyla were mainly composed of microeukaryotic phytoplankton, with various relative
abundances between different lakes (Fig. 2a). The Chrysophyta (35.04% in
CH~76.69 % in XH), Chlorophyta (13.94% in KT~35.37% in YY), and Cryptophyta
(0.01% in YO~23.73% in CH) were most abundant in lakes. The Cryptophyta in KT
was significantly more abundant than in XH and YO, and Alveolate in KT was
significantly more abundant than in YO ($P<0.05$) (Fig. 2b). Meanwhile, it was noticed
that the relative abundances of some phyla varied between lakes but not significantly,
with Chytridiomycota, Cercozoa, and Cryptophyta in XH being higher than those in
YO. The Arthropoda represented 0.10% in CH~4.11% in YO; Alveolate represented
0.03% in XH~1.01% in CH, and unclassified SAR represented 1.07 % in XH~5.27 %
in YO.
The relative abundance of the dominant taxa in the same lake had some
interannual variation. The Chrysophyta in CH_19, YO_19, and YY_19 samples were
lower than the other samples, while the Cryptophyta in CH_19 and YY_19 were
lower than the other samples in CH, YY, respectively. The proportion of Arthropoda
in YO_19 reached 70.09%, which was remarkably higher than the different samples
(Fig. 2a).
A total of 24 dominant genera were identified (Fig. 2c), accounting for 81.22%
sequences in CH, 79.43% in KT, 61.22% in XH, 65.95% in YO, and 59.06% in YY.
The dominant genera were mainly *Hydrurus*, *Paraphysomonas*, *Ochromonas*, and
*Monochrysis* belonging to Chrysophyta, *Komma* in Cryptophyta, *Monomastix*,
*Chlamydomonas*, and *Raphidonema* in Chlorophyta.
As shown in Fig. 2c, the abundance of the dominant genera differed among the
lakes investigated interannually. The relative abundance of *Komma* varied from 0 to
48.49%, which showed an increasing trend over the year in CH and YY. The ranges of
*Paraphysomonas* and *Ochromonas* were 0.28~41.98% and 0.22~15.82%, showing an
increase followed by a decrease in XH and YO over the year. *The Hydrurus* in XH_18
and YO_18 was higher than the other samples. *Raphidonema* was significantly more





abundant in CH than in other lakes. *Chrysosphaerell* and *Synura* in KT, except for
compared with CH, were significantly higher than other lakes ($P<0.05$, Table S2).

The indices (OTUs, Shannon index, and NTI) had interannual variation but

showed no significance ($P>0.05$) (Fig. 3a, c, e). The order of NTI and Shannon all
showed 2018>2017>2019; OTUs were highest in the expedition season 2017 and
lowest in 2019. The ranges of OTUs and Shannon index were 151~244 and 2.06~3.26,
respectively, with YY having the lowest value and was significantly lower than CH
and KT (Fig. 3b, d, $P<0.05$). The range of NTI was 0.80~1.42, with the lowest value
in YO and significantly lower than KT (Fig. 3f, $P<0.05$). KT had the highest Shannon
and NTI, while CH had the highest number of OTUs (Fig. 3b, d, f).

The total number of OTUs shared in 2017-2019 was 276, and the unique OTUs

were 31 (2017), 34 (2018), 70 (2019) (Fig. 3g). The Venn diagram showed that the
total number of OTUs shared by the five lakes was 129, and the unique OTUs were
62 (CH),37 (KT),5 (XH),14 (YO), and 14 (YY) (Fig. 3h).

The NMDS results divided the samples into five clusters according to their

similarity of microeukaryotic community (stress value = 0.14) (Fig. 4a). In addition,
the analysis of similarity (ANOSIM) based on Bray-Curtis distance indicated that the
differences between lakes were significant (Global R = 0.613, $P<0.01$). Meanwhile,
no significant differences were detected by ANOSIM among interannual variations
(R=0.013, $P$=0.393).

UPGMA clustering analysis (Fig.4b) showed the same lakes in a different year,

such as CH_17 and CH_18, YY_17 and YY_18, YO_17 and YO_18 clustered into
one clade, respectively. For other lakes, KT_18 and XH_18 clustered as one clade,
CH_19, and YY_19 clustered as one clade, CH_17, and XH_19 clustered as one clade.
YO lake was distant from other lakes and clustered into a separate one.
3.3 Driving factors and co-occurrence patterns

Canonical correspondence analysis (CCA) demonstrated that the first two

sequencing axes explained 16.7% and 15.5% of community variation (Fig. 5a). The





samples from the same lake were closer, with a more similar community structure.
More importantly, the Monte Carlo analysis confirmed that only the water
temperature significantly affected the microeukaryotic community ($P<0.01$). The
variation partition analysis (VPA) indicated that environmental factors monitored
explained 14.19% of microeukaryotic community variability among lakes and still
had a large amount of unexplained community variation (85.8%, Fig. 5b).
A total of 223 nodes linked by 1941 edges was made up microeukaryotic network.
The majority of nodes in the network had many connections. Notably, the positive
associations among species were predominant in the network (Fig. 5c), with 82.25%,
whereas the portion of negative association was only 17.75%. In addition, the positive
interactions were mainly within the same taxonomic affiliations, such as Chrysophyta,
or between a few different taxonomic affiliations, such as Chrysophyta and
Chlorophyta. While the negative correlations mainly were reflected between
Chrysophyta and Chlorophyta. We found that only about 8% of OTUs directly
correlated with environmental factors. Meanwhile, only four of the top 20 OTUs with
the highest degree centrality were directly associated with environmental factors (WT,
$PO_4$-P), and three belonged to Chrysophyta and one to Cercozoa.
24 nodes were identified as potential keystone species (Table S3), which
contained *Heteromita* belonging to Cercozoa, seven genera belonging to Chrysophyta,
such as *Spumella*, *Ochromonas*, and *Chromulina*. The Chlorophyta keystone genera
included *Chloromonas* and *Chlamydomonas*, and other genera were from
Bacillariophyta and Alveolata.
3.4 Community assembly processes
The Sloan neutral community model (NCM) showed the importance of
stochastic processes for microeukaryotic communities (Fig. 6a), with the neutral
processes explaining 56.8% community variation. In addition, the Sloan neutral
model classified microeukaryotic taxa into three groups (above prediction, below
prediction, and neutral prediction). We found that the neutral group (within 95%





confidence interval), with richness and abundance ratios of 79.7% and 90.4%,
respectively, were both much higher than the above and below prediction groups,
which was dominated by Chrysophyta, Chlorophyta, and Cryptophyta (Fig. 6b, c).
The above prediction group accounted for 8.8% of the microeukaryotic richness but
corresponded to only 0.75% of the abundance, dominated by Chrysophyta,
Chlorophyta, and Chytridiomycota. Cryptophyta accounted for 13.3% of the
abundance in the neutral group but was almost absent in the other two groups. In
contrast, Chytridiomycota was present in 10.4% abundance in the two groups
mentioned above, but only 0.1% in the neutral group.
The variation of βNTI ranged from -1.65~1.31 with a mean value of -0.48 (Fig.
6d), which was mainly distributed in the region of stochastic processes and supported
the results of the neutral model. The community assembly process analysis showed
that stochastic rather than deterministic processes controlled the community assembly.
Among them, homogenizing dispersal dominated, with a proportion of 64.76%,
followed by undominated process and dispersal limitation, with 32.38% and 2.86%,
respectively (Fig. 6e).
**4 Discussion**
4.1 Diversity and dominant taxa
The environmental conditions (e.g., low light and low nutrient, etc.) in Antarctic
freshwater lakes differed from temperate lakes. These special features and relative
isolation result in unique communities and the survival strategies of the species
adapted to such conditions (Pearce 2008). In our study, the survival of taxa
(Chrysophyta, Chlorophyta, and Cryptophyta) might depend on their survival
strategies to withstand harsh conditions, which made them as the predominant species.
Chrysophyta dominated in five lakes examined in our study, including *Hydrurus*,
*Paraphysomonas*, *Ochromonas*, and *Monochrysis*. Firstly, the dominance may be due
to the adaptation to low nutrient availability; the relatively high surface to volume
ratio contributes to the uptake of nutrients at low concentrations, Which have been





reported in high latitude polar lakes (Charvet et al., 2012) . Secondly, Chrysophyta
still keeps a high proportion under low light conditions, as they can adapt to changing
light conditions (Yubuki et al., 2008). Furthermore, Chrysophyta is mixotrophic and
even can swim, which allows them to get available nutrients from other
microorganisms, reducing the need for dissolved nutrients in the water (F R Pick and
Lean 1984; Katechakis and Stibor 2006). In addition, when the environmental
conditions change dramatically, such as freezing and nutrient changes, Chrysophyta
can form cysts (Nicholls 1995), protecting cells from resisting an unsuitable
environment. All these aspects make Chrysophyta has the advantage to be the
predominant species in the five Antarctic lakes.

Chlorophyta was the second most dominant taxon in our study (13.94%~

35.37%), containing mainly *Monomastix*, *Chlamydomonas*, and *Raphidonema*.
Chlorophyta is typically represented by flagellated species such as *Chlamydomonas*
spp., which dominate the phytoplankton in different trophic statuses and respond to
adverse environmental conditions by forming temporary groups (Allende and
Mataloni 2013; Izaguirre et al., 2003; Toro et al., 2007). Several unicellular algae can
mix acid fermentation, and some obligate photoautotrophic species respond by photo-
acclimation processes involving the accumulation of chlorophyll to increase the light
capture efficiency (Atteia et al., 2013; Morgan-Kiss et al., 2016). These characteristics
might partially be explained how Chlorophyta survived and occupied a specific
advantage of the important reason in lakes we studied.

Cryptophyta was the third dominant taxon observed. Indeed, their dominance has

been interpreted as evidence of heterotrophic growth in winter and mixotrophic
throughout the year (Unrein et al., 2014). Cryptophyta dominates under perennially
ice-covered and coastal saline lakes in continental Antarctica. The ingestion of
bacteria by mixotrophic Cryptophyta has been observed in two perennially ice-
covered lakes (Fryxell and Hoare) in the McMurdo Dry Valleys (Roberts and
Laybourn-Parry 1999).

Compared with other aquatic ecosystems (Sun et al., 2021), the diversity of

microeukaryotes in Antarctic lakes was significantly lower (Shannon 2.06~3.26,
OTUs 151~244). The diversity of microorganisms reported decreases from mid-
latitude to the poles (Santos et al., 2020). The isolation and harsh conditions,
especially the lower temperatures and nutrients, prevailing in Antarctic lakes account
for a low microeukaryotic diversity. In addition, the species-area relationships model
(SAR) states that increased species number with increasing habitat area within a
specific area (Ma 2018). An increase in the ice-free area drastically modifies
biodiversity (Duffy et al., 2017; Lee et al., 2017; Pertierra et al., 2017). Our results
supported the SAR model, observing more diversity and richness in CH and KT,
where habitat areas were much larger than the YY and YO.
**4.2 Influence of environmental factors on the community**

Previous great efforts have demonstrated that abiotic factors affect microbial

diversity and community composition (Quiroga et al., 2013; Sun et al., 2017). Our
study found that only water temperature was a significant driving factor for
community change among the abiotic factors analyzed. Some microorganisms have
evolved to grow under a defined temperature, allowing differences in temperature
adaptation of different species (Wilkins et al., 2013). Water temperature has become a
major driving factor for changes in microeukaryotic communities by regulating
cellular activity and metabolic rates (Margesin and Miteva 2011). The retreat of
glaciers due to global warming had the risk of reducing the abundance and diversity
of microorganisms, and more attention should be paid to the impact of water
temperature changes on community structure (Garcia-Rodriguez et al., 2021). The
water temperature in the YO lakes was significantly higher than in the other lakes, and
YO clustered into a separate clade (Fig. 5b).

Nevertheless, a small amount of community variation could be explained by

measured environmental variables in our analysis. This indicated that these
environmental factors played a minor role in shaping microeukaryotic community
structure. There were many unexplained variations (Fig. 5), and some possible causes



have been indicated. Firstly, the nonconsecutive of environmental factors among
different expedition seasons was deficient in our study. There are also many vital
abiotic factors in Antarctic freshwater lakes, including solar cycle, light availability,
ice cover (thickness and duration), physical changes as snow melts and mixes, and
hydrological changes (Allende and Izaguirre 2003; Lizotte 2008). Secondly, the
relationship between microorganisms (symbiotic or competitive) cannot be quantified,
which is an essential factor influencing community structure. Predation pressure
manifests itself in lakes as a top-down control of microeukaryotes (Blomqvist 1997).
Thirdly, stochastic processes such as ecological drift (birth, death) may cause
unexplained community variation (Zhang et al., 2018).
## 4.3 Co-occurrence patterns and keystone taxa
Network analysis can help us understand complex biological interactions and
ecological rules for community assembly within a specific ecological niche (Li and
Hu 2021; Lupatini et al., 2014). Microorganisms form various ecological relationships,
ranging from mutualism to competition, ultimately shaping microbial abundances
(Faust and Raes 2012). Positive associations in a network often indicate common
preferred environmental conditions or niche-overlapping, whereas negative
associations mean competition or niche division (Faust and Raes 2012). By analyzing
the network, we found that the positive correlations were much more than the
negative correlations in the co-occurrence network (87% vs. 13%), indicating that
species coexistence was achieved mainly by symbiotic relationships between species.
In addition, only 8% OTUs were significantly correlated with environmental factors,
suggesting that microeukaryotes have a relatively lower response to environmental
factors and these could weaken the role of environment selection in community
assembly. Previous studies have shown the high response of microeukaryotic
communities to mid-and late-stage diatom blooms promotes deterministic processes
(Hou et al., 2020).
In co-occurrence networks, keystone species play a critical role in maintaining



the structure and function of the microbial community, and the loss of essential
species may lead to the fracturing of networks (Zhang et al., 2022). The keystone
species in this study belonged mainly to Chlorophyta, Chrysophyta, Bacillariphyta,
and Cercozoa. *Heteromita* has significant genetic variation and promotes bacterial
degradation of alkylbenzenes through predation (Ekelund et al., 2004). *Spumella* is a
heterotrophic microorganism commonly found in freshwater and soil (Boenigk et al.,
2005). As mixed trophic organisms, *Ochromonas* prey on bacteria and are, therefore, a
critical link between bacteria and higher trophic levels (Andersson et al., 1989).
*Chloromonas* has motile trophic cells that can grow in the snow to give it a green
color and, together with Chlamydomonas, are thought to have a strong carbon
concentration mechanism (Hu 1998).
4.4 Community assembly processes

In general, deterministic and stochastic processes exist simultaneously in the

community assembly (Chase 2010; He et al., 2021). Several factors such as habitat
connectivity and size (Orrock and Watling 2010), productivity (Chase 2010),
disturbance (Liang et al., 2020), predation (Chase et al., 2009), and resource
availability (Kardol et al., 2013) influence the relative importance of stochastic and
deterministic processes in the community assembly. The importance of stochastic
processes has been previously illustrated for other microeukaryotic communities from
the aquatic ecosystem (Chen et al., 2019; Wang et al., 2020). In our study, the results
supported the prominent role of stochastic processes in shaping the microeukaryotic
community assembly than deterministic processes. Hence, environmental variables
explained only a small number of variations in our study's microeukaryotic
community, and a small number of taxa were significantly correlated with
environmental factors.

In our study, the microeukaryotic community showed a good fit (57%) to the

neutral model (Fig. 6a), which suggested community variation can be explained by
stochastic processes such as birth, death, and migration to a large extent. The NCM

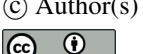



462 can also attribute the observed patterns of community assembly to different

463 population behavior (Zhang et al., 2021b). The NCM separated taxa into three groups,

464 and these groups were different in community structure (Fig. 6), similar to the result

465 of microeukaryotic community in the channel (Zhang et al., 2021b), indicating these

466 taxa might differ in their adaptability to the environment or dispersal rate (Chen et al.,

467 2019). In addition, the neutral group contributed a high proportion to both abundance

468 (90.4%) and richness (79.7%) in our study. In a subtropical river, the neutral group

469 also dominates the microeukaryotic community in terms of richness and abundance

470 (Chen et al., 2019). Similar results in this study suggested that microeukaryotes in this

471 study were more susceptible to stochastic processes.

472  Furthermore, the null model results showed that stochastic processes (mainly

473 homogenizing dispersal and undominated process) dominated the community

474 assembly (Fig. 6c). The importance of stochastic processes has been previously

475 illustrated by the microeukaryotic communities of lakes in East Antarctica (Logares et

476 al., 2018). Abrupt changes in environmental conditions can affect the relative

477 contribution of community assembly processes. For example, increasing the nutrients

478 and regulating ecological scheduling (Chan et al., 2002; Jiang and Patel 2008; Liu et

479 al., 2019), perennial fertilization in the soil (Liang et al., 2020), and the activities of

480 long-term cultivation of rice fields (Liu et al., 2020b) all have cause changes in the

481 relative contribution of stochastic and deterministic processes. It has been believed

482 that if changing environmental factors are not significant or do not force selection on

483 species, stochastic processes still dominate (Zhou et al., 2014). The extreme

484 environmental conditions over a long period might lessen the ecological selection

485 pressure on microeukaryotes. Furthermore, the explanation for the dominance of the

486 stochastic process might also be due to the long-term adaptation of species to the

487 environment, which leads to a low response, as also confirmed by the fungal

488 community assembly (Powell et al., 2015).

489  Our study's microeukaryotic community tended to homogenize during dispersal,

490 and the community compositions were relatively stable. Antarctic freshwater lakes





can receive external microbial colonies by the input of microorganisms from the
surrounding ice melt, atmospheric transport, human activities, or bird migration
(Unrein et al., 2005). Water bodies have been reported occupied with a high
proportion of homogenizing dispersal (Zeng et al., 2019). Most microorganisms
detected in the sea also have been found present in lakes in East Antarctica, pointing
to that some marine taxa in the lake may be the product of homogenizing dispersal
from the ocean to the lake (Logares et al., 2018). In addition, the lakes were covered
in ice for most of the year and were limited by geographical distance, resulting in the
dispersal limitation of microorganisms (2.86%). Undominant processes accounted for
32.38% of community assembly in our study, including ecological drift and other
complex processes that have not been fully quantified, such as weak selection and
diffusion (Mo et al., 2018a), suggesting that microeukaryotic communities might be
formed by some highly complex assembly mechanisms in Antarctic freshwater lakes.

**5 Conclusion**

In conclusion, the microeukaryotic community was dominated by phytoplankton,
mainly Chrysophyta, Chlorophyta, and Cryptophyta, with spatial and temporal
variation in the relative abundance of dominant taxa from five freshwater lakes on the
Fildes Peninsula, Antarctic. This study highlighted the first time the importance of
stochastic processes and co-occurrence patterns in shaping the microeukaryotic
community of this area. The environmental variables explained only about 30% of the
community variation. Microbial interactions were predominantly symbiotic,
indicating common preferred environmental conditions or niche-overlapping.
Stochastic processes played a very prominent role in microeukaryotic community
assembly, and the low response to environmental factors might enhance the
proportion of stochastic processes. Our study provides a better understanding of the
dynamic patterns and ecological processes of microeukaryotic community structure in
Antarctic oligotrophic lakes (Fildes Peninsula).

**Data Availability Statement**

The raw 18S reads have been deposited into the NCBI Sequence Read Archive



database with the accession numbers of SRP359325.

## Author Contribution Statement

Conceptualization: Chunmei Zhang and Yonghong Bi. Methodology: Chunmei
Zhang and Wei Luo. Molecular technique:Huirong Li. Physico-chemical properties:
Bin Wang, Yangjie Li, and Zhongqiang Ji. Sample collection:Yinxin Zeng and
Haitao Ding. Funding acquisition: Yonghong Bi and Wei Luo. Supervision: Yonghong
Bi and Wei Luo. Writing - original draft: Chunmei Zhang. Writing - review & editing:
Yonghong Bi and Wei Luo.

## Competing interests

The authors declare that they have no known competing financial interests or
personal relationships that could have appeared to influence the work reported in this
paper.

## Acknowledgments

We have special thanks to the 34th, 35th, 36th Chinese Antarctic Research
Expedition (CHINARE) and the Antarctic Great Wall Ecology National Observation
and Research Station (PRIC) for their strong logistic supports of this field survey in
the summer season of 2017/2018, 2018/2019, 2019/2020. Field Samples were
approved by the Chinese Arctic and Antarctic Administration (CAA). This research
was supported by the National Natural Science Foundation of China (No. 91851201;
No.31971477).

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





**Figure Captions**
**Fig. 1** (a) Location of the five studied lakes in Fildes Peninsula, King George Island,
Antarctica, (b) Lake Xi Hu (XH), (c) Lake Yan Ou (YO), (d) Lake Chang Hu (CH), (e)
Lake Yue Ya (YY), (f) Lake Kitec (KT).
**Fig.2** Comparison of microeukaryotic community composition. (a)Temporal and
spatial dynamics of relative abundance on Phylum level in five lakes from 2017 to
2019. 17, 18, and 19 expedition season of 2017/2018, 2018/2019, and 2019/2020,
respectively. (b) Differential analysis of microeukaryotes in different lakes. Lakes that
showed no significant differences were not shown($P>0.05$). (c)Temporal and spatial
dynamics of relative abundance on Genus level in five lakes. (Note: $**P < 0.01$, $*P <$
$0.05$). (Chlorophyta: *Aphelida*, *Atractomorpha*, *Chlamydomonas*, *Chloromonas*,
*Chlorothrix*, *Choricystis*, *Crustomastix*, *Microglen*, *Monomastix*, *Nannochloris*,
*Raphidonema*; Chrysophyta: *Chrysosphaerell*, *Hydrurus*, *Mallomonas*, *Monochrysis*,
*Ochromonas*, *Paraphysomonas*, *Spumella*, *Synura*, *Tessellaria*; Cryptophyta: *Komma*;
Haptophyta: *Diacronema*; Pyrrophyta: *Heterocapsa*; Glissomonadida: *Heteromita*.
The relative abundance at any lake was less than 1% was defined as others).
**Fig.3** Microbial diversity and Venn diagram in different years and lakes. (a, b)
variations in microbial OTUs; (c, d) variations in microbial Shannon index; (e, f)
variations in within-community nearest-taxon index (NTI); (g, h) Venn diagram
showing the unique and shared operational taxonomic units (OTUs). Homogeneity
and one-way ANOVA analysis of variance was used to test the indices' significance.
"ns" represents no significant differences (*P*>0.05). Significant differences (*P*<0.05)
are indicated by different alphabet letters.
**Fig. 4** Temporal variability analysis of Non-metric multidimensional scaling (NMDS)
ordination of microeukaryotic communities (a) and clustering of five lakes based on
similarity (b).





**Fig.5** The effect of environmental variables on microeukaryotic communities, and co-
occurrence pattern. Canonical correlation analysis plots(a) and variance partitioning
analysis (b), respectively. Sal: salinity; WT: water temperature; $NO_2$ -N: nitrite
nitrogen; $PO_4$ -P: phosphate phosphorus. ** $P < 0.01$. (c) Networks analysis revealing
the interspecies associations between microeukaryotic OTUs, and the correlation
between environmental factors and OTUs in lakes integrated networks. The size of
each OTUs or environmental factor (node) is proportional to the degree centrality.
Others: other phyla and unclassified taxa.
**Fig.6** Relative influences of deterministic and stochastic processes on
microeukaryotic community assembly based on the neutral community model (NCM)
and the null model. (a) Fit of the neutral community model (NCM) of community
assembly. Nm indicates the metacommunity size, and $R^2$ indicates the fit to the
neutral model. Neutral prediction is within 95% confidence interval (black), while
non-neutral processes include above and below prediction (dark green and red). (b)
Proportions in richness and abundance of the three groups (above prediction, below
prediction, and neutral prediction) based on the NCM. (c) Composition of the three
groups in abundance for microeukaryotic community. (d) β-nearest-taxon index
(βNTI) range of community. (e) Null model analysis revealing the fraction of
ecological processes. The percent of community assembly is governed primarily by
various deterministic processes, including homogenous and heterogeneous selections
and stochastic processes, including dispersal limitations and homogenizing dispersal
and undominated processes (mainly ecological drift).





**Fig. 1**

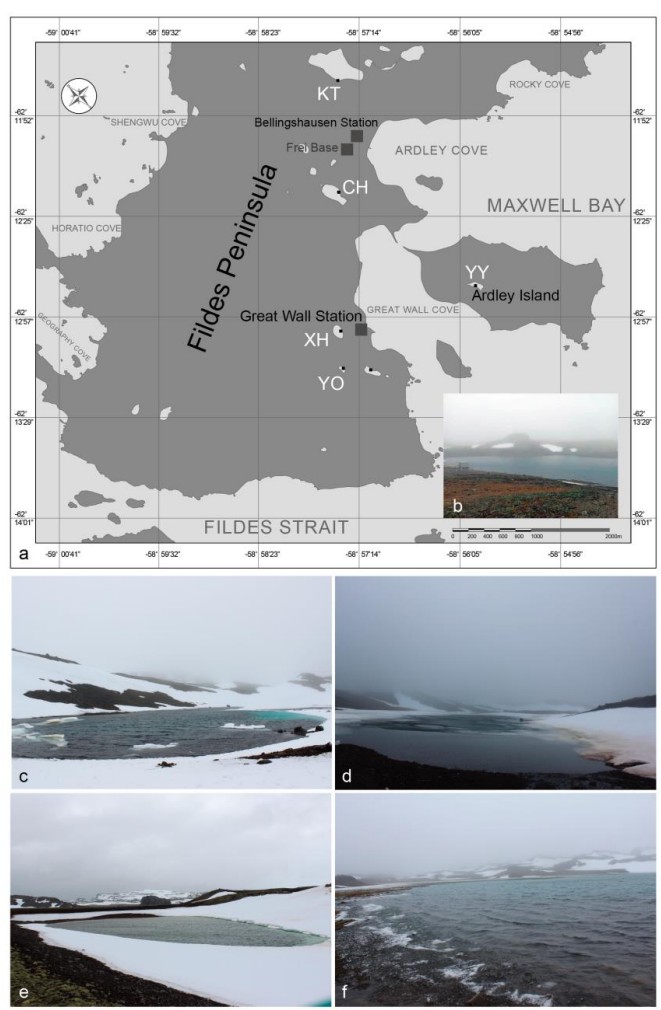



**Fig. 1** (a) Location of the five studied lakes in Fildes Peninsula, King George Island,
Antarctica, (b) Lake Xi Hu (XH), (c) Lake Yan Ou (YO), (d) Lake Chang Hu (CH), (e)
Lake Yue Ya (YY), (f) Lake Kitec (KT).





**Fig. 2**

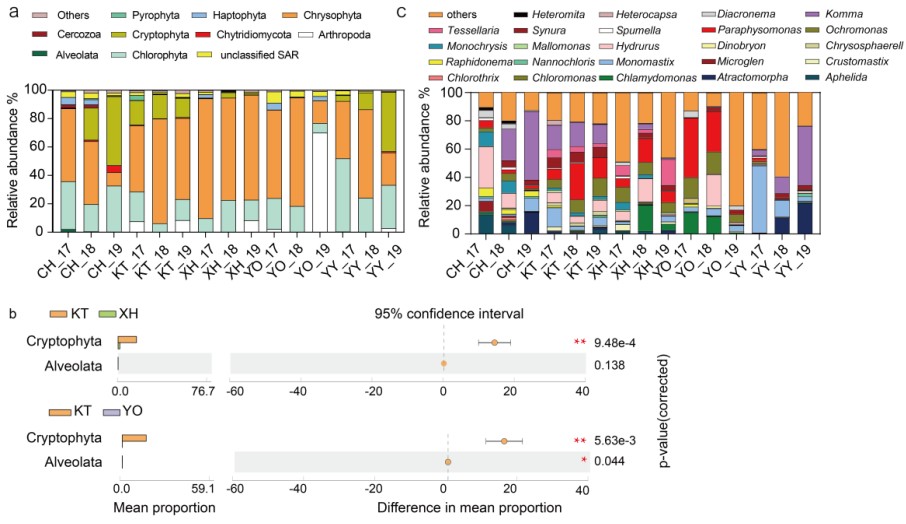



**Fig.2** Comparison of microeukaryotic community composition. (a)Temporal and
spatial dynamics of relative abundance on phylum level in five lakes from 2017 to
2019. 17, 18, and 19 expedition season of 2017/2018, 2018/2019, and 2019/2020,
respectively. (b) Differential analysis of microeukaryotes in different lakes. Lakes that
showed no significant differences were not shown($P>0.05$). (c)Temporal and spatial
dynamics of relative abundance on genus level in five lakes. (Note: **$P < 0.01$, *$P <$
0.05). (Chlorophyta: *Aphelida*, *Atractomorpha*, *Chlamydomonas*, *Chloromonas*,
*Chlorothrix*, *Choricystis*, *Crustomastix*, *Microglen*, *Monomastix*, *Nannochloris*,
*Raphidonema*; Chrysophyta: *Chrysosphaerell*, *Hydrurus*, *Mallomonas*, *Monochrysis*,
*Ochromonas*, *Paraphysomonas*, *Spumella*, *Synura*, *Tessellaria*; Cryptophyta: *Komma*;
Haptophyta: *Diacronema*; Pyrrophyta: *Heterocapsa*; Glissomonadida: *Heteromita*.
The relative abundance at any lake was less than 1% was defined as others).



**Fig. 3**

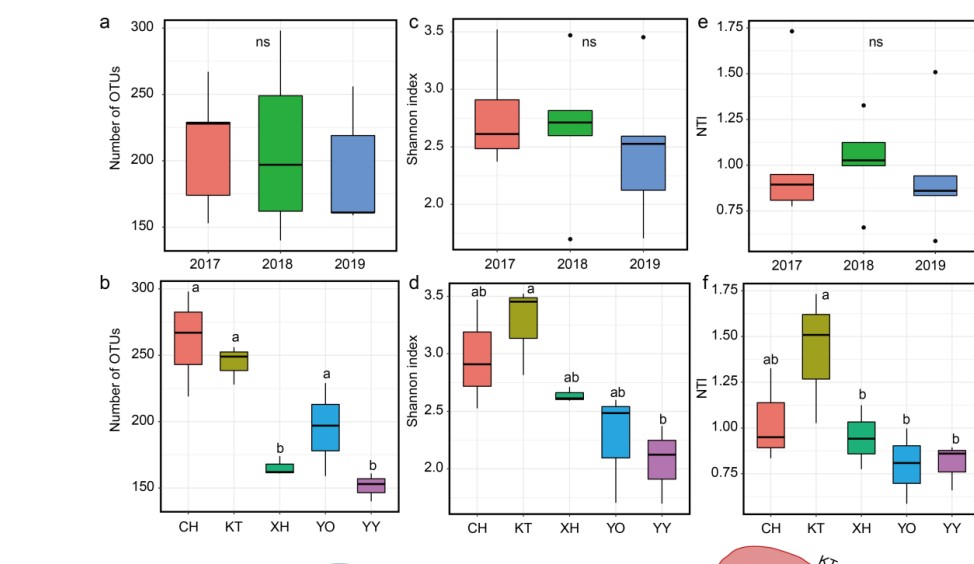



**Fig.3** Microbial diversity and Venn diagram in different years and lakes. (a, b)
variations in microbial OTUs. (c, d) variations in microbial Shannon index. (e, f)
variations in within-community nearest-taxon index (NTI). (g, h) Venn diagram
showing the unique and shared operational taxonomic units (OTUs). Homogeneity
and one-way ANOVA analysis of variance was used to test the indices' significance.
"ns" represents no significant differences ($P>0.05$). Significant differences ($P<0.05$)
are indicated by different alphabet letters.


**Fig. 4**

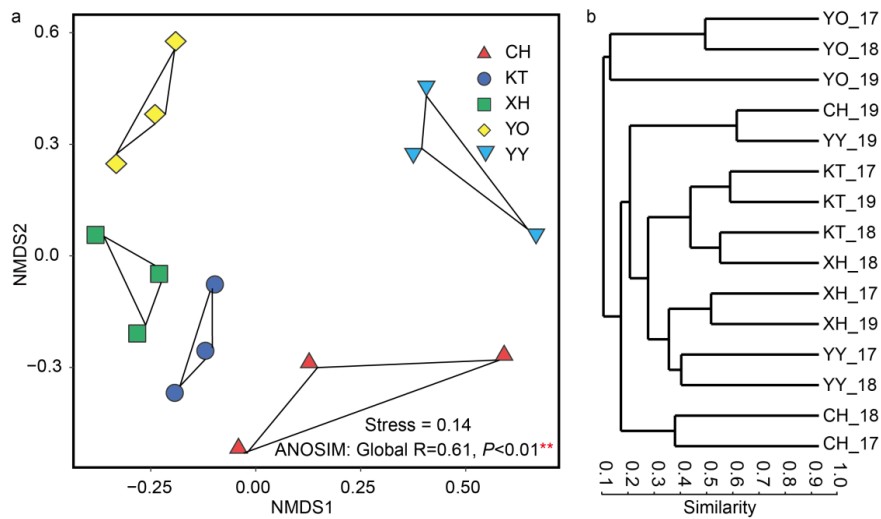


**Fig. 4** Temporal variability analysis of Non-metric multidimensional scaling (NMDS)
ordination of microeukaryotic communities (a) and clustering of five lakes based on
similarity (b).



**Fig. 5**

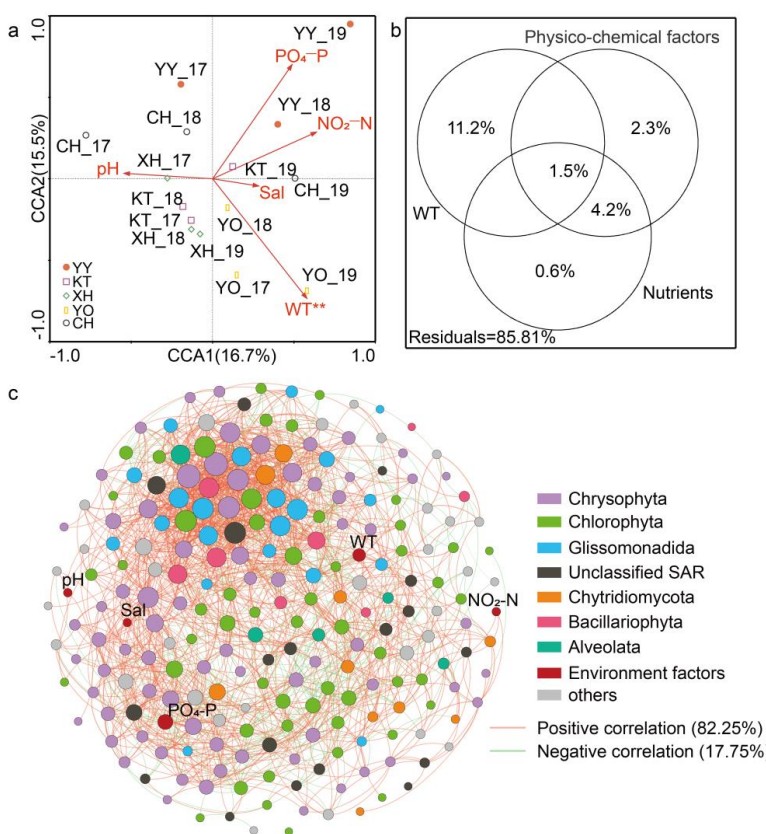


**Fig.5** The effect of environmental variables on microeukaryotic communities, and co-
occurrence pattern.  Canonical correlation analysis plots(a) and variance partitioning
analysis (b), respectively. ** $P < 0.01$. (c) Networks analysis revealing the
interspecies associations between microeukaryotic OTUs, and the correlation between
environmental factors and OTUs in lakes integrated networks. The size of each OTUs
or environmental factor (node) is proportional to the degree centrality. Others: other
phyla and unclassified taxa.




**Fig. 6**

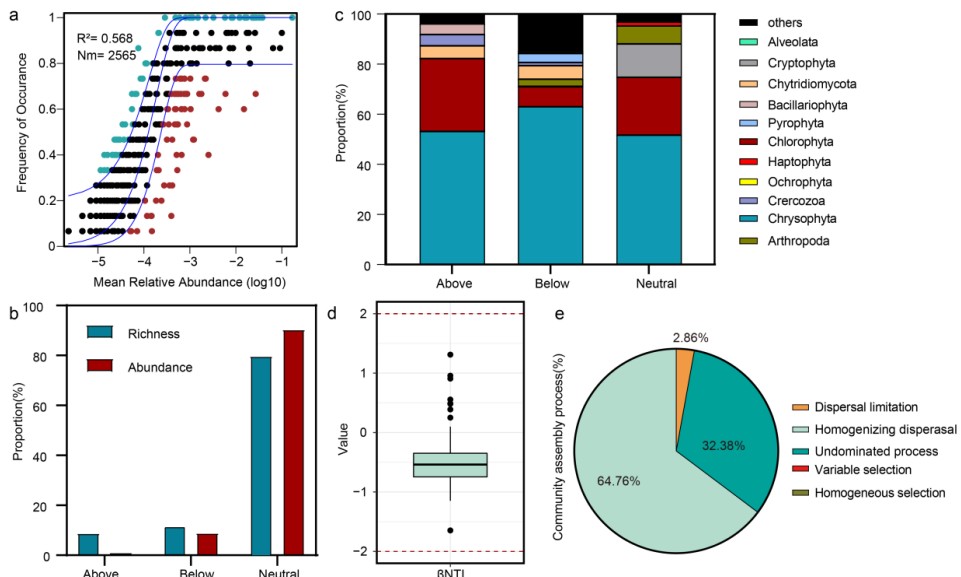


**Fig.6** Relative influences of deterministic and stochastic processes on
microeukaryotic community assembly based on the neutral community model (NCM)
and the null model. (a) Fit of the neutral community model (NCM) of community
assembly. Nm indicates the metacommunity size, and $R^2$ indicates the fit to the
neutral model. Neutral prediction is within 95% confidence interval (black), while
non-neutral processes include above and below prediction (dark green and red). (b)
Proportions in richness and abundance of the three groups (above prediction, below
prediction, and neutral prediction) based on the NCM. (c) Composition of the three
groups in abundance for microeukaryotic community. (d) β-nearest-taxon index
(βNTI) range of community. (e) Null model analysis revealing the fraction of
ecological processes. The percent of community assembly is governed primarily by
various deterministic processes, including homogenous and heterogeneous selections
and stochastic processes, including dispersal limitations and homogenizing dispersal
and undominated processes (mainly ecological drift).