# Peer review of "Diversity and assembly processes of microbial eukaryotic"

_Biogeosciences, 2022_

## Author Comment (AC1)

**Revierwer1**

Microeukaryotes are wide distribution and play importance role in aquatic ecosystem. Diversity and assembly processes of microeukaryotes should be paid more attention. Antarctic freshwater lakes are a kind of non-pollution freshwater ecosystem and Microeukaryotes are important contributor of primary producer. Diversity and assemblage of microeukaryotes would give us a clue to get insight into the whole ecosystem. Five freshwater lakes' microeukaryotic communities on the Fildes Peninsula were screened in this manuscript, sample data were provided, diversity, co-occurrence patterns and assembly processes have been analyzed. In my opinion, this work is very interesting and provided a better understanding of the dynamic patterns and ecological processes of microeukaryotic community structure in oligotrophic lakes. I recommend it is worthy to be published after minor revision.

**General Response to Reviewer 1 Comments**

Thanks a lot for your comments and suggestions. We are very appreciated with your helpful advice and have made our efforts to revise the manuscript with clarifications/elaborations as following.

**Our response** is in **normal font** and colored in **blue**, and the *revised text* is in italic font and colored in **blue**.

**General comments**

1. As parameter of diversity, Shannon index should be mentioned in the abstract.

**Response:** Thanks for your advice. Revised as "Alpha diversity varied among lakes, with…." to "*Richness (113~268) and Shannon index (1.70~3.50) varied among lakes, with…*" in the abstract part.

2. WT and PO₄-P should be mentioned in the abstract, but not the "Environmental factors".

**Response:** Agreed with you. We have revised in the abstract. "Environmental variables only explained 39% of the variation in community structure, *with the water temperature and orthophosphate being identified as the important driving factors (P<0.05)*."

3. In the discussion, dominant taxa and keystone species were mentioned, is there any difference between the two? If there is no difference, please use the same one.

**Response:** There are differences between the dominant taxa and keystone species.

Dominant taxa are considered absolutely dominant (abundant), determined by their relative abundance, which are regarded to have significant control on community structure. The keystone species may be abundant or rare, not depending on their relative abundance, and their disappearance or weakening is thought to lead to the fracturing of microbial community networks (Banerjee et al., 2018).

In our study, the dominant taxa and keystone species are not identical and are therefore discussed separately.

(1) Banerjee, S., Schlaeppi, K. and van der Heijden, M.G.A. (2018). Keystone taxa as drivers of microbiome structure and functioning. Nature Reviews Microbiology, 16(9), 567-576. https://doi.org/10.1038/s41579-018-0024-1.

4. The conclusion is too long and should be shortened.

**Response:** Shortened as "*The unique microbial eukaryotic community structure and low alpha diversity (richness and Shannon index) was demonstrated in five freshwater lakes on the Fildes Peninsula, Antarctic. The importance of stochastic processes and co-occurrence patterns in shaping the microbial eukaryotic community of this area was proved. Water temperature and orthophosphate were identified as important driving factors for driving variation of community structure (P<0.05). Stochastic processes played a prominent role in community assembly. This study provides a better understanding of the dynamic patterns and assembly processes of microbial eukaryotic*

*community structure in Antarctic oligotrophic lakes (Fildes Peninsula).*"

---

## Author Response (AR1)

Dear editor,

We are deeply grateful for the efforts of you and reviewers to improve the quality of our manuscript. We have made our efforts to revise the manuscript with clarifications/elaborations as following.

In this version, we have made major revisions based on the comments: (1) revised the comment on prefiltering, diatoms, bioinformatic choices and the relationships in co-occurrence network analysis, (2) re-analyzed the data by removing the metazoan sequences in advance of the analysis and revised results, figures, discussion, and more references though the main results and conclusion of our study did not be affected and (3) provided more detailed information about the material and method, (4) removed the description about keystone species. A list of all the changes made can be found in the point-by-point response to the reviewers' comments.

**Our response** is in **normal font** and colored in **blue**, and *the revised text* is in *italic font* and colored in **blue**. Lines number refer to the track changed version.

**Editor**

In the revised version please be sure to address the comments that are specified within the reviews, as you did in your response to reviews. I point this out as a few are addressed within the response but not highlighted where it will be in the text. In particular, the difference between dominant and keystone species (as you discussed for Reviewer AC1), comments on prefiltering and diatoms by AC2 and justification of bioinformatic choices made.

**Response:** We have revised in our manuscript and highlighted where it will be in the text.

**About the difference between dominant and keystone species:** Dominant taxa always have high abundance relative to other species in a community, and have proportionate effects on environmental conditions, community diversity, and/or ecosystem function (Avolio et al., 2019). However, the keystone species may be abundant or rare, not depending on their relative abundance, and their disappearance or weakening is thought to lead to the fracturing of microbial community networks (Banerjee et al., 2018).

We agree with the reviewers' suggestions. In our manuscript, we calculated which species might be potential keystone species in the co-occurrence network, but there

are currently no more data to demonstrate how these species play key roles in the microbial eukaryotic community. Therefore, we decide to remove the description of keystone species from the manuscript.

(1) Banerjee, S., Schlaeppi, K. and van der Heijden, M.G.A. (2018). Keystone taxa as drivers of microbiome structure and functioning. Nature Reviews Microbiology, 16(9), 567-576. https://doi.org/10.1038 /s41579-018-0024-1.

(2) Avolio, M.L., Forrestel, E.J., Chang, C.C., La Pierre, K.J., Burghardt, K.T. and Smith, M.D. (2019) Demystifying dominant species. New Phytol 223(3), 1106-1126. https://doi.org/10.1111/nph.15789.

**About Prefiltering:** Microbial eukaryotes (0.2~20 μm, pico-/nano-eukaryotes) constitute important components in microbial food webs and play an important role in the biogeochemical cycles (Grob et al., 2007; Massana et al., 2015; Unrein et al., 2014), as well as contributing to plankton biomass and carbon export (Hernandez-Ruiz et al., 2018; Leblanc et al., 2018). However, the microbial eukaryotes have been neglected for a long term due to their small cell size and lack of conspicuous morphological features. The molecular approach can be used to compensate for the lack of traditional microscopic methods, providing us with a convenient way to study these small-sized eukaryotes. The application of 18S rRNA gene-based molecular tools has revealed high taxonomic diversity of microbial eukaryotes in some oligotrophic and extreme regions (Marquardt et al., 2016; Richards et al., 2005; Zhao et al., 2011). Nevertheless, research studies focused on exploring the molecular diversity and the population fluctuations in these far cold and oligotrophic Antarctic lakes are limited.

The appearance of large metazoan as multicellular organisms could cause an artificially underestimate the smaller size organisms "mostly unicellular organisms" in the molecular sequencing. So proper prefiltering is necessary for understanding these particular populations as nano- and pico-, specifically does the high-throughput sequencing. This is why we choose the prefiltering for the high-throughput sequencing for the very beginning of these consecutive yearly investigations.

The phrase "microeukaryotes" might be the problematic issue, so we would amend it into "microbial eukaryotes" in this case, including the size ≤20 μm pico- and nano-eukaryotes. Thus, we do hope that our study would provide a better understanding of the dynamic patterns and ecological processes of microbial eukaryotic community structure in Antarctic oligotrophic lakes (Fildes Peninsula).

We have revised in the introduction. "*Microbial eukaryotes (0.2~20 μm,*

*pico-/nanoeukaryotes) constitute important components in microbial food webs and play an important role in the biogeochemical cycles (Grob et al., 2007; Massana et al., 2015; Unrein et al., 2014), as well as contributing to plankton biomass and carbon export (Hernandez-Ruiz et al., 2018; Leblanc et al., 2018). However, the microbial eukaryotes have been neglected for a long term due to their small cell size and lack of conspicuous morphological features. The molecular approach can be used to compensate for the lack of traditional microscopic methods, providing us with a convenient way to study these small-sized eukaryotes. The application of 18S rRNA gene-based molecular tools has revealed high taxonomic diversity of microbial eukaryotes in some oligotrophic and extreme regions (Marquardt et al., 2016; Richards et al., 2005; Zhao et al., 2011). Nevertheless, research studies focused on exploring the molecular diversity and the population fluctuations in these far cold and oligotrophic Antarctica lakes are limited.*" (L61-76)

(1) Grob, C., Ulloa, O., Li, W.K.W., Alarcon, G., Fukasawa, M. and Watanabe, S. (2007) Picoplankton abundance and biomass across the eastern South Pacific Ocean along latitude 32.5 degrees S. Marine Ecology Progress Series 332, 53-62. https://doi.org/10.3354/meps332053.

(2) Massana, R., Gobet, A., Audic, S., Bass, D., Bittner, L., Boutte, C., Chambouvet, A., Christen, R., Claverie, J.M., Decelle, J., Dolan, J.R., Dunthorn, M., Edvardsen, B., Forn, I., Forster, D., Guillou, L., Jaillon, O., Kooistra, W.H.C.F., Logares, R., Mahe, F., Not, F., Ogata, H., Pawlowski, J., Pernice, M.C., Probert, I., Romac, S., Richards, T., Santini, S., Shalchian-Tabrizi, K., Siano, R., Simon, N., Stoeck, T., Vaulot, D., Zingone, A. and de Vargas, C. (2015) Marine protist diversity in European coastal waters and sediments as revealed by high-throughput sequencing. Environmental Microbiology 17(10), 4035-4049. https://doi.org/10.1111/1462-2920.12955.

(3) Unrein, F., Gasol, J.M., Not, F., Forn, I. and Massana, R. (2014) Mixotrophic haptophytes are key bacterial grazers in oligotrophic coastal waters. Isme Journal 8(1), 164-176. https://doi.org/10.1038/ismej.2013.132.

(4) Hernandez-Ruiz, M., Barber-Lluch, E., Prieto, A., Alvarez-Salgado, X.A., Logares, R. and Teira, E. (2018) Seasonal succession of small planktonic eukaryotes inhabiting surface waters of a coastal upwelling system. Environ Microbiol 20(8), 2955-2973. https://doi.org/10.1111/1462-2920.14313.

(5) Leblanc, K., Queguiner, B., Diaz, F., Cornet, V., Michel-Rodriguez, M., Durrieu de Madron, X., Bowler, C., Malviya, S., Thyssen, M., Gregori, G., Rembauville, M., Grosso, O., Poulain, J., de Vargas, C., Pujo-Pay, M. and Conan, P. (2018) Nanoplanktonic diatoms are globally overlooked but play a role in spring blooms and carbon export. Nat Commun 9(1), 953. https://doi.org/10.1038/s41467-018-03376-9.

(6) Marquardt, M., Vader, A., Stubner, E.I., Reigstad, M. and Gabrielsen, T.M. (2016) Strong Seasonality of Marine Microbial Eukaryotes in a High-Arctic Fjord (Isfjorden, in West Spitsbergen, Norway). Appl Environ Microbiol 82(6), 1868-1880. https://doi.org/10.1128/AEM.03208-15.

(7) Richards, T.A., Vepritskiy, A.A., Gouliamova, D.E. and Nierzwicki-Bauer, S.A. (2005) The

molecular diversity of freshwater picoeukaryotes from an oligotrophic lake reveals diverse, distinctive and globally dispersed lineages. Environ Microbiol 7(9), 1413-1425. https://doi.org/10.1111/j.1462-2920.2005.00828.x.

(8)   Zhao, B., Chen, M., Sun, Y., Yang, J. and Chen, F. (2011) Genetic diversity of picoeukaryotes in eight lakes differing in trophic status. Can J Microbiol 57(2), 115-126. https://doi.org/10.1139/w10-107.

**About the relative abundance of diatom:** Just as for the response to reviewer 2, we do not deny the ecological status of diatoms in Antarctic lakes. However, this study was based on a high-throughput sequencing approach with a more precise means of focusing on the smaller size of eukaryotes (0.2~20 μm) that are easily underestimated, which is a complement to previous studies. We have already added to the study of diatoms in the introduction. "*Based on microscopic observation, diatoms in the lakes of Fildes Peninsula region accounted for 59.8% of the total number of phytoplankton species, and the water temperature and nutrition resulted in the variation of nano-and microalgae abundance and composition in lakes (Zhu et al., 2010).*" (L84-88)

(1)   Zhu, G.H., OHTANI Shuji, HU Chuan-yu, HE Jian-feng, JIN Mao, YU Pei-song and Jian-ming, P. (2010) Impact of global climate change on antarctic freshwater algae. China Environmental Science 30(3), 400-404.

I would also ask that you carefully review the comprehensive points highlighted by reviewer AC2. They rightfully point out that you are using a Qiime1 (a 7 years old release) instead of Qiime2, which limits your analysis to OTUs instead of ASVs. Deblur and DADA2 are very much the current standard for the sorts of analysis that you are undertaking, whether you treat your data set at the ASV or a higher ranking (so OTUs picked from ASVs) and that is independent of using QIIME or other bioinformatic tools. Like reviewer AC2, I feel justification in the text is needed for this choice. For example, ASV picking in certain cases does result in sequencing depth loss but not always and so I would suggest that you provide evidence of this statement with your data set. Again – nothing wrong with OTUs but dismissing ASVs off hand is not in line with current approaches in the field.

**Response:** We are very grateful to the editor as well as the reviewers for their suggestions. As mentioned by the editor, both analyses (OTUs and ASVs) have their own characteristics, which have been discussed in previous studies (Amos et al., 2020; Glassman and Martiny 2018). Furthermore, we never deny either analysis method.

The analysis of OTUs obtained using the UPARSE clustering method have still been widely used for high-throughput sequencing analysis (Gad et al., 2022; Reboul et al., 2021; Sun et al., 2022; Xu et al., 2022; Zhang et al., 2022). Some of the diversity indices in our study are more comparable to previous similar researches using 97% sequence similarity OTUs (Chen et al., 2022; Wang et al., 2021; Wang et al., 2020b), as comparisons of these statistics using the same bioinformatics tool still seem to remain persuasive, but the broad-scale ecological patterns remained robust regardless of the feature-clustering method (Glassman and Martiny 2018; Li et al., 2019).

In addition, after preliminary analysis of the data based on ASVs, we found that compared to OTUs, the total number of sequences was reduced by 45% after ASVs picking and the number of normalized and rarified sequences was reduced by 35% (16717 vs. 10894).

We believe that the analysis of OTUs is appropriate for our current study and can also clearly describe our results. We have noted studies based on ASVs and do not exclude subsequent studies will use this approach. Most importantly, we never deny either analysis method.

We have revised in our manuscript. "*The analysis to OTUs obtained using the UPARSE clustering method have still been widely used for high-throughput sequencing analysis (Gad et al., 2022; Reboul et al., 2021; Sun et al., 2022; Xu et al., 2022; Zhang et al., 2022). Some of the diversity indices in our study were more comparable to previous similar researches using 97% sequence similarity OTUs (Chen et al., 2022; Wang et al., 2021; Wang et al., 2020b), as comparisons of these statistics using the same bioinformatics tool still seem to remain persuasive (Glassman and Martiny 2018; Li et al., 2019).*" (L196-202)

(1) Amos, G.C.A., Logan, A., Anwar, S., Fritzsche, M., Mate, R., Bleazard, T. and Rijpkema, S. (2020) Developing standards for the microbiome field. Microbiome 8(1). https://doi.org/10.1186/s40168-020-00856-3.

(2) Glassman, S.I. and Martiny, J.B.H. (2018) Broadscale Ecological Patterns Are Robust to Use of Exact Sequence Variants versus Operational Taxonomic Units. Ecological and Evolutionary Science 3(4), e00148-00118. https://doi.org/10.1128/mSphere.

(3) Gad, M., Hou, L., Cao, M., Adyari, B., Zhang, L., Qin, D., Yu, C.P., Sun, Q. and Hu, A. (2022) Tracking microeukaryotic footprint in a peri-urban watershed, China through machine-learning approaches. Science of the Total Environment 806(Pt 1), 150401. https://doi.org/10.1016/j.scitotenv.2021.150401.

(4) Reboul, G., Moreira, D., Annenkova, N.V., Bertolino, P., Vershinin, K.E. and Lopez-Garcia, P.

(2021) Marine signature taxa and core microbial community stability along latitudinal and vertical gradients in sediments of the deepest freshwater lake. Isme Journal 15(11), 3412-3417. https://doi.org/10.1038/s41396-021-01011-y.

(5) Sun, P., Wang, Y., Huang, X., Huang, B.Q. and Wang, L. (2022) Water masses and their associated temperature and cross-domain biotic factors co-shape upwelling microbial communities. Water research 215. https://doi.org/10.1016/j.watres.2022.118274.

(6) Xu, D., Kong, H., Yang, E.J., Wang, Y., Li, X., Sun, P., Jiao, N., Lee, Y., Jung, J. and Cho, K.H. (2022) Spatial dynamics of active microeukaryotes along a latitudinal gradient: Diversity, assembly process, and co-occurrence relationships. Environ Res 212(Pt A), 113234. https://doi.org/10.1016/j.envres.2022.113234.

(7) Zhang, W., Wan, W., Lin, H., Pan, X., Lin, L. and Yang, Y. (2022) Nitrogen rather than phosphorus driving the biogeographic patterns of abundant bacterial taxa in a eutrophic plateau lake. Science of the Total Environment 806(Pt 4), 150947. https://doi.org/10.1016/j.scitotenv.2021.150947.

(8) Chen, Z., Gu, T., Wang, X., Wu, X. and Sun, J. (2022) Oxygen gradients shape the unique structure of picoeukaryotic communities in the Bay of Bengal. Science of the Total Environment 814, 152862. https://doi.org/10.1016/j.scitotenv.2021.152862.

(9) Wang, F., Huang, B., Xie, Y., Cai, S., Wang, X. and Mu, J. (2021) Diversity, Composition, and Activities of Nano- and Pico-Eukaryotes in the Northern South China Sea With Influences of Kuroshio Intrusion. Frontiers in Marine Science 8. https://doi.org/10.3389/fmars.2021.658233.

(10) Wang, Y., Li, G., Shi, F., Dong, J., Gentekaki, E., Zou, S., Zhu, P., Zhang, X. and Gong, J. (2020) Taxonomic Diversity of Pico-/Nanoeukaryotes Is Related to Dissolved Oxygen and Productivity, but Functional Composition Is Shaped by Limiting Nutrients in Eutrophic Coastal Oceans. Front Microbiol 11, 601037. https://doi.org/10.3389/fmicb.2020.601037.

(11) Li, Y., Gao, Y., Zhang, W., Wang, C., Wang, P., Niu, L. and Wu, H. (2019) Homogeneous selection dominates the microbial community assembly in the sediment of the Three Gorges Reservoir. Science of the Total Environment 690, 50-60. https://doi.org/10.1016/j.scitotenv.2019.07.014.

I note that reviewer AC2 asked which chemistry was used for your MiSeq run and that was not included but should be in the modified version.

**Response:** We have provided more information regarding the sequencing. "*The PCR product was extracted from 2% agarose gel and purified using the AxyPrep DNA Gel Extraction Kit (Axygen Biosciences, Union City, CA, USA) according to manufacturer's instructions and quantified using Quantus™ Fluorometer (Promega, USA).*" (L173-176)

**Revierwer1**

Microeukaryotes are wide distribution and play importance role in aquatic ecosystem. Diversity and assembly processes of microeukaryotes should be paid more

attention. Antarctic freshwater lakes are a kind of non-pollution freshwater ecosystem and Microeukaryotes are important contributor of primary producer. Diversity and assemblage of microeukaryotes would give us a clue to get insight into the whole ecosystem. Five freshwater lakes' microeukaryotic communities on the Fildes Peninsula were screened in this manuscript, sample data were provided, diversity, co-occurrence patterns and assembly processes have been analyzed. In my opinion, this work is very interesting and provided a better understanding of the dynamic patterns and ecological processes of microeukaryotic community structure in oligotrophic lakes. I recommend it is worthy to be published after minor revision.

**General Response to Reviewer 1 Comments**

Thanks a lot for your comments and suggestions. We are very appreciated with your helpful advice and have made our efforts to revise the manuscript with clarifications/elaborations as following.

**Our response** is in **normal font** and colored in **blue**, and the *revised text* is in italic font and colored in **blue**. Lines number refer to the track changed version.

**General comments**

1. As parameter of diversity, Shannon index should be mentioned in the abstract.

**Response:** Thanks for your advice. Revised as "Alpha diversity varied among lakes, with…." to "*Richness (113~268) and Shannon index (1.70~3.50) varied among lakes, with…*" in the abstract part. (L24-26)

2. WT and $PO_4^{3-}$ should be mentioned in the abstract, but not the "Environmental factors".

**Response:** Agreed with you. We have revised in the abstract. "Environmental variables explained 39% of the variation in community structure, *with the water temperature and phosphate being identified as the important driving factors (P<0.05).*" (L31-32)

3. In the discussion, dominant taxa and keystone species were mentioned, is there any difference between the two? If there is no difference, please use the same one.

**Response:** Dominant taxa always have high abundance relative to other species in a community, and have proportionate effects on environmental conditions, community diversity, and/or ecosystem function (Avolio et al., 2019). However, The keystone species may be abundant or rare, not depending on their relative abundance, and their disappearance or weakening is thought to lead to the fracturing of microbial community networks (Banerjee et al., 2018).

We agree with the reviewers' suggestions. In our manuscript, we calculated which species might be potential keystone species in the co-occurrence network, but there are currently no more data to demonstrate how these species play key roles in the microbial eukaryotic community. Therefore, we decide to remove the description of keystone species from our manuscript.

(1) Banerjee, S., Schlaeppi, K. and van der Heijden, M.G.A. (2018). Keystone taxa as drivers of microbiome structure and functioning. Nature Reviews Microbiology, 16(9), 567-576. https://doi.org/10.1038/s41579-018-0024-1.

(2) Avolio, M.L., Forrestel, E.J., Chang, C.C., La Pierre, K.J., Burghardt, K.T. and Smith, M.D. (2019) Demystifying dominant species. New Phytol 223(3), 1106-1126. https://doi.org/10.1111/nph.15789.

4. The conclusion is too long and should be shortened.

**Response:** Shortened as "*The unique microbial eukaryotic community structure and low alpha diversity (richness and Shannon index) were demonstrated in five freshwater lakes on the Fildes Peninsula, Antarctica. The importance of stochastic processes and co-occurrence patterns in shaping the microbial eukaryotic community of this area was proved. Water temperature and phosphate were identified as important driving factors for driving variation of community structure (P<0.05). Stochastic processes played a prominent role in community assembly. This study provided a better understanding of the dynamic patterns and assembly processes of microbial eukaryotic community structure in Antarctic oligotrophic lakes (Fildes Peninsula).*" (L610-625)

**Revierwer2**

The article "Diversity and assembly processes of microeukaryotic community in

Fildes Peninsula Lakes (West Antarctica)" by authors Zhang et al. is an impressive effort to characterize and interpret protist communities in hard to reach and understudied ecosystems. The authors analyze protist communities from the same lakes every austral summer for three years. This provides a unique opportunity to understand how stable these communities are over time. The authors discuss the dominant taxa—Crysophyta, Cryptophyta, and Chlorophyta—and how their abundance relates to environmental factors and is influenced by biotic interactions, and whether the community assembly processes are mainly deterministic or stochastic. Overall, the authors conclude that environmental factors contribute little to community composition, interactions are mainly positive between taxa, and stochastic factors primarily shape community assembly. This is a unique study, due to its temporal component, and it documents important aspects of rapidly changing ecosystems in Antarctica. Below are comments that I believe will help improve the manuscript.

**General Response to Reviewer 2 Comments**

Thanks for your comments and suggestions. We are appreciated with your helpful advice, and we have made our efforts to revise the manuscript with clarifications/elaborations as following.
**Our response** is in **normal font** and colored in **blue**, and *the revised text* is in *italic font* and colored in ***blue***. Lines number refer to the track changed version.

**General comments**

My main concern is the pre-filtering step in the methods and how that might influence the subsequent results and interpretation. The methods state that the water was pre -filtered through 20 micron mesh-size to remove "mesoplankton and large particles" and then biomass was collected onto a 0.2 micron pore-size filter. This step actually removes all of the microplankton and leaves behind the nano and picoplankton. This has obvious implications for the title and the language throughout the manuscript, but also has more important implications for the interpretation of the results. The authors note that there is less diversity in these samples than in similar studies, which I suspect could be due to more aggressive filtering? The authors also note in the introduction that diatoms have been studied in Antarctic lakes previously but they do not report finding significant proportions of diatoms in their samples,

which could also be an artifact of the size fractionation in this study. Finally, the authors report mainly positive relationships in their co-occurrence network analysis. Again, I think this may be due to the size selection, as microeukaryotes are more likely to graze nano and pico eukaryotes. However, the observation that there seems to be more niche-overlap than competition between nano and pico eukaryotes remains very interesting. Lastly, I feel that it is incorrect to refer to the positive interactions as symbiotic without further evidence documenting symbiotic relationships between the node OTUs being discussed.

**About Prefiltering?**

**Response:** Microbial eukaryotes (0.2~20 μm, pico-/nano-eukaryotes) constitute important components in microbial food webs and play an important role in the biogeochemical cycles (Grob et al., 2007; Massana et al., 2015; Unrein et al., 2014), as well as in plankton biomass and contribute to carbon export (Hernandez-Ruiz et al., 2018; Leblanc et al., 2018). However, the microbial eukaryotes have been neglected for a long term due to their small cell size and lack of conspicuous morphological features. The molecular approach can be used to compensate for the lack of traditional microscopic methods, providing us with a convenient way to study these small-sized eukaryotes. The application of 18S rRNA gene-based molecular tools has revealed high taxonomic diversity of microbial eukaryotes in some oligotrophic and extreme regions (Marquardt et al., 2016; Richards et al., 2005; Zhao et al., 2011). Nevertheless, research studies focused on exploring the molecular diversity and the population fluctuations in these far cold and oligotrophic Antarctic lakes are limited.

The appearance of large metazoan as multicellular organisms could cause an artificially underestimate the smaller size organisms "mostly unicellular organisms" in the molecular sequencing. So proper prefiltering is necessary for understanding these particular populations as nano- and pico-, specifically does the high-throughput sequencing. This is why we choose the prefiltering for the high-throughput sequencing for the very beginning of these consecutive yearly investigations.

The phrase **"microeukaryotes"** might be the problematic issue, so we would amend it into "microbial eukaryotes" in this case, including the size ≤20 μm pico- and nanoeukaryotes. Thus, we do hope that our study would provide a better understanding of the dynamic patterns and ecological processes of microbial

eukaryotic community structure in Antarctic oligotrophic lakes (Fildes Peninsula).

We have revised in our manuscript. "*Microbial eukaryotes (0.2~20 μm, pico-/nanoeukaryotes) constitute important components in microbial food webs and play an important role in the biogeochemical cycles (Grob et al., 2007; Massana et al., 2015; Unrein et al., 2014), as well as contributing to plankton biomass and carbon export (Hernandez-Ruiz et al., 2018; Leblanc et al., 2018). However, the microbial eukaryotes have been neglected for a long term due to their small cell size and lack of conspicuous morphological features. The molecular approach can be used to compensate for the lack of traditional microscopic methods, providing us with a convenient way to study these small-sized eukaryotes. The application of 18S rRNA gene-based molecular tools has revealed high taxonomic diversity of microbial eukaryotes in some oligotrophic and extreme regions (Marquardt et al., 2016; Richards et al., 2005; Zhao et al., 2011). Nevertheless, research studies focused on exploring the molecular diversity and the population fluctuations in these far cold and oligotrophic Antarctica lakes are limited.*" (L61-76)

(1) Grob, C., Ulloa, O., Li, W.K.W., Alarcon, G., Fukasawa, M. and Watanabe, S. (2007). Picoplankton abundance and biomass across the eastern South Pacific Ocean along latitude 32.5 degrees S. Marine Ecology Progress Series, 332, 53-62. https://doi.org/DOI 10.3354/meps332053.

(2) Massana, R., Gobet, A., Audic, S., Bass, D., Bittner, L., Boutte, C., Chambouvet, A., Christen, R., Claverie, J.M., Decelle, J., Dolan, J.R., Dunthorn, M., Edvardsen, B., Forn, I., Forster, D., Guillou, L., Jaillon, O., Kooistra, W.H.C.F., Logares, R., Mahe, F., Not, F., Ogata, H., Pawlowski, J., Pernice, M.C., Probert, I., Romac, S., Richards, T., Santini, S., Shalchian-Tabrizi, K., Siano, R., Simon, N., Stoeck, T., Vaulot, D., Zingone, A. and de Vargas, C. (2015). Marine protist diversity in European coastal waters and sediments as revealed by high-throughput sequencing. Environmental Microbiology, 17(10), 4035-4049. https://doi.org/10.1111/1462-2920.12955.

(3) Unrein, F., Gasol, J.M., Not, F., Forn, I. and Massana, R. (2014). Mixotrophic haptophytes are key bacterial grazers in oligotrophic coastal waters. Isme Journal, 8(1), 164-176. https://doi.org/10.1038/ismej.2013.132.

(4) Hernandez-Ruiz, M., Barber-Lluch, E., Prieto, A., Alvarez-Salgado, X.A., Logares, R. and Teira, E. (2018). Seasonal succession of small planktonic eukaryotes inhabiting surface waters of a coastal upwelling system. Environ Microbiol, 20(8), 2955-2973. https://doi.org/10.1111/1462-2920.14313.

(5) Leblanc, K., Queguiner, B., Diaz, F., Cornet, V., Michel-Rodriguez, M., Durrieu de Madron, X., Bowler, C., Malviya, S., Thyssen, M., Gregori, G., Rembauville, M., Grosso, O., Poulain, J., de Vargas, C., Pujo-Pay, M. and Conan, P. (2018) Nanoplanktonic diatoms are globally overlooked but play a role in spring blooms and carbon export. Nat Commun 9(1), 953. https://doi.org/10.1038/s41467-018-03376-9.

(6) Marquardt, M., Vader, A., Stubner, E.I., Reigstad, M. and Gabrielsen, T.M. (2016). Strong Seasonality of Marine Microbial Eukaryotes in a High-Arctic Fjord (Isfjorden, in West

Spitsbergen, Norway). Appl Environ Microbiol, 82(6), 1868-1880. https://doi.org/10.1128/AEM.03208-15.

(7) Richards, T.A., Vepritskiy, A.A., Gouliamova, D.E. and Nierzwicki-Bauer, S.A. (2005). The molecular diversity of freshwater picoeukaryotes from an oligotrophic lake reveals diverse, distinctive and globally dispersed lineages. Environ Microbiol, 7(9), 1413-1425. https://doi.org/10.1111/j.1462-2920.2005.00828.x.

(8) Zhao, B., Chen, M., Sun, Y., Yang, J. and Chen, F. (2011). Genetic diversity of picoeukaryotes in eight lakes differing in trophic status. Can J Microbiol, 57(2), 115-126. https://doi.org/10.1139/w10-107.

**About the relative abundance of diatom?**

**Response:** Diatoms in the lakes of Fildes Peninsula region were reported as the first predominant population, accounting for 59.8% of the total number of phytoplankton species (Zhu et al., 2010). Similarly, our microscopic observations on the phytoplankton samples (without pre-filtering) showed that diatoms are dominant as well (unpublished data), which was not involved into this discussion yet. Still, we have traced some sequencing reads related to diatoms in the sequencing dataset even after the "artificial" prefiltering.

However, this study was based on a high-throughput sequencing approach with a more precise means of focusing on the smaller size of eukaryotes that are easily underestimated in such Antarctic oligotrophic lakes. In this case, diatoms were not the dominant taxa (relative abundance >1% at any lake) in the 0.2~20 μm size range, and their relative abundance varied from 0.007% in YY_19 ~ 0.633% in CH_18. Diatoms were not abundant within the small-eukaryotes (0.2~20 μm) community in agreement with other studies (Hernandez-Ruiz et al., 2018).

We have already added to the study of diatoms in the introduction. "*Based on microscopic observation, diatoms in the lakes of Fildes Peninsula region accounted for 59.8% of the total number of phytoplankton species, and the water temperature and nutrients resulted in the variation of nano-and microalgae abundance and composition in lakes (Zhu et al., 2010).*" (L84-88)

(1) Zhu, G.H., OHTANI Shuji, HU Chuan-yu, HE Jian-feng, JIN Mao, YU Pei-song and Jian-ming, P. (2010). Impact of global climate change on antarctic freshwater algae. China Environmental Science, 30(3), 400-404.

(2) Hernandez-Ruiz, M., Barber-Lluch, E., Prieto, A., Alvarez-Salgado, X.A., Logares, R. and Teira, E. (2018). Seasonal succession of small planktonic eukaryotes inhabiting surface waters of a coastal upwelling system. Environ Microbiol, 20(8), 2955-2973. https://doi.org/10.1111/1462-2920.14313.

**The lower diversity compared with other similar studies?**

**Response:** The results of the comparison with other similar studies were shown in Table 1. The results showed that the richness (OTUs) and Shannon index of microbial eukaryotes were indeed lower in our study area. More references have been added to our manuscript.

Table 1 Comparison of microbial eukaryotes richness and Shannon index in different study areas.

|  | Object of study | Richness | Range of richness | Shannon index |
|---|---|---|---|---|
| Our study | Microbial eukaryotes (0.2~20μm) | 520 | 113~268 (178±46) | 1.70~3.50 (2.72±0.52) |
| The Northern South China Sea **(Wang et al., 2021)** | Pico-/Nanoeukaryotes | 3198/3233 | / | / |
| The Coastal Oceans **(Wang et al., 2020b)** | Pico-/Nanoeukaryotes | 1590 | 178~233 | / |
| The surface waters of a coastal upwelling system **(Hernandez-Ruiz et al., 2018)** | Small eukaryotes (0.2~20μm) | / | 180~511 (337±89) | 2.37~5.18 (4.12±0.84) |

**Note:** The "/" indicates no relevant data in the References.

(1) Wang, F., Huang, B., Xie, Y., Cai, S., Wang, X. and Mu, J. (2021). Diversity, Composition, and Activities of Nano- and Pico-Eukaryotes in the Northern South China Sea With Influences of Kuroshio Intrusion. Frontiers in Marine Science, 8. https://doi.org/10.3389/fmars.2021.658233.

(2) Wang, Y., Li, G., Shi, F., Dong, J., Gentekaki, E., Zou, S., Zhu, P., Zhang, X. and Gong, J. (2020). Taxonomic Diversity of Pico-/Nanoeukaryotes Is Related to Dissolved Oxygen and Productivity, but Functional Composition Is Shaped by Limiting Nutrients in Eutrophic Coastal Oceans. Front Microbiol, 11, 601037. https://doi.org/10.3389/fmicb.2020.601037.

(3) Hernandez-Ruiz, M., Barber-Lluch, E., Prieto, A., Alvarez-Salgado, X.A., Logares, R. and Teira, E. (2018). Seasonal succession of small planktonic eukaryotes inhabiting surface waters of a coastal upwelling system. Environ Microbiol, 20(8), 2955-2973. https://doi.org/10.1111/1462-2920.14313.

**Co-occurrence network?**

**Response:** We agreed with the reviewer that the information regarding the positive interactions as symbiotic without further evidence is not suitable. Co-occurrence/non-coexistence patterns among populations may reflect either niche

overlap/partitioning or positive/negative ecological interactions such as commensalism, mutualism, or competition. In our study, we found many positive correlations in the co-occurrence network. However, further studies are necessary to corroborate the biological interactions and other nonrandom processes (for example, cross-feeding versus niche overlap) between species pairs detected by network analyses. Consequently, we have revised in the abstract and discussion.

**Abstract:** *"Finally, network analysis revealed comprehensive co-occurrence relationships in the microbial eukaryotic community (positive correlation 82% vs. negative correlation 18%)."* (L34-35)

**Discussion:** **"**￼*By analyzing the network, we found that the positive correlations were much more than the negative correlations in the co-occurrence network (82% vs. 18%), revealing that positive relationships (e.g., due to cross-feeding, niche overlap, mutualism, and/or commensalism) might exhibit a more important role than negative relationships (e.g., predator-prey relationships, host-parasite relationships and/or competition)* (Chen and Wen 2021) *in studied Antarctic lake ecosystem. A similar result has been found in small planktonic eukaryotes (0.2~20 μm) inhabiting the surface waters of a coastal upwelling system (Hernandez-Ruiz et al., 2018). Notwithstanding, further studies are necessary to corroborate the biological interactions and other nonrandom processes (for example, cross-feeding versus niche overlap) between species pairs detected by network analyses.***"** (L515-524)

(1) Chen, W. and Wen, D. (2021). Archaeal and bacterial communities assembly and co-occurrence networks in subtropical mangrove sediments under Spartina alterniflora invasion. Environ Microbiome, 16(1), 10. https://doi.org/10.1186/s40793-021-00377-y.

(2) Hernandez-Ruiz, M., Barber-Lluch, E., Prieto, A., Alvarez-Salgado, X.A., Logares, R. and Teira, E. (2018). Seasonal succession of small planktonic eukaryotes inhabiting surface waters of a coastal upwelling system. Environ Microbiol, 20(8), 2955-2973. https://doi.org/10.1111/1462-2920.14313.

**Technical comments:**

(1) In the methods and throughout, I suggest choosing one spelling for each lake and sticking with it for consistency.

**Response:** Thank you. We have revised one spelling for each lake. When first described, the five lakes were described as Lake Xi Hu (XH), Lake Yan Ou (YO), Lake Chang Hu (CH), Lake Yue Ya (YY), and Lake Kitec (KT). In other parts, each lake was indicated by the abbreviation.

(2) Line 123- I suggest abbreviating temperature just as Temp, similar to using Sal for salinity. WT adds an unnecessary additional acronym. And please complete the statement YSI Model 30 … what type of instrument, a CTD?

**Response:** We used the WT to describe the "water temperature" based on other references, and the temp may be misinterpreted as air temperature. In the descriptions of results, figures, and tables in our manuscript, we have consistently used WT.
And we have revised the "YSI Model 30" to "*RBRconcerto C.T.D (Canada)*".

(3) Line 143- "PCR products were pooled and purified using the DNA gel extraction kit." I think this statement is a mistake as pooling should not occur at this step …

**Response:** We have revised a detailed description to the PCR amplification. "*The PCR product was extracted from 2% agarose gel and purified using the AxyPrep DNA Gel Extraction Kit (Axygen Biosciences, Union City, CA, USA) according to manufacturer's instructions and quantified using Quantus™ Fluorometer (Promega, USA). Purified amplicons were pooled in equimolar and paired-end sequenced on an Illumina MiSeq PE300 platform (Illumina, San Diego, USA) according to the standard protocols by Wefind Biotechnology Co., Ltd. (Wuhan, China).*" (L173-179)

(4) Line 148- please provide more information regarding the sequencing. The first line of the section states the instrument model, but which version chemistry was used? How many base pairs were sequenced (300?)? And paired or single end?

**Response:** Thanks for your suggestion. We have provided more information regarding the sequencing as the response mentioned above. "*The PCR product was extracted from 2% agarose gel and purified using the AxyPrep DNA Gel Extraction Kit (Axygen Biosciences, Union City, CA, USA) according to manufacturer's instructions and quantified using Quantus™ Fluorometer (Promega, USA). Purified amplicons were pooled in equimolar and paired-end sequenced (2×300) on an Illumina MiSeq platform (Illumina, San Diego,USA) according to the standard protocols by Wefind Biotechnology Co., Ltd. (Wuhan, China).*" (L173-179)

(5) Line 149- the bioinformatics methods are a bit dated. For instance, why did you use qiime instead of qiime 2? Why OTUs instead of ASVs? Likewise, the SILVA database used is not the most recent and you might also consider using the PR2 database, which is curated specifically for protists. To be clear, I do not necessarily recommend redoing the analysis with more up to date methodology, but I do

recommend justifying your decisions with an explanatory sentence.

**Response:** Thanks for your question. I am very sorry for the trouble I caused to the reviewer by my typing mistakes. For example, the "QIIME" should be "*QIIME1.9.1*", and the "SILVA database (Release 132)" should be "*SILVA database (Release 138)*". Furthermore, the data were analyzed with the free online Majorbio I-Sanger Cloud Platform (http://www.majorbio.com/). In this platform, the latest version of all the analysis software would be updated from time to time. And we utilized the SILVA database (Release 138), which contains high-quality 18S genes (Quast et al. 2013), to determine operational taxonomic units (OTUs).

In the manuscript, we have made a detailed addition to the Illumina MiSeq sequencing and Processing of sequencing data. "*The raw 18S rRNA gene sequencing reads were demultiplexed, quality-filtered by fastp version 0.20.0 (Chen et al., 2018) and merged by FLASH version 1.2.7 (Magoc and Salzberg 2011) with the following criteria: (i) the 300 bp reads were truncated at any site receiving an average quality score of <20 over a 50 bp sliding window, and the truncated reads shorter than 50 bp were discarded, reads containing ambiguous characters were also discarded; (ii) only overlapping sequences longer than 10 bp were assembled according to their overlapped sequence. The maximum mismatch ratio of overlap region is 0.2. Reads that could not be assembled were discarded; (iii) Samples were distinguished according to the barcode and primers, and the sequence direction was adjusted, exact barcode matching, 2 nucleotide mismatch in primer matching.*"

"*Operational taxonomic units (OTUs) with 97% similarity cutoff were clustered using UPARSE version 7.1 (Edgar 2013), and chimeric sequences were identified and removed. The taxonomy of each OTU representative sequence was analyzed by RDP Classifier version 2.2 (Wang et al., 2007) against the 18S rRNA database (Silva v138) using confidence threshold of 0.7 (Quast C et al., 2013).*" (L180-195)

(1) Quast C, Pruesse E, Yilmaz P, Gerken J, Schweer T, Yarza P, PepliesJ and FO, G. (2013). The SILVA ribosomal RNA gene database project: improved data processing and web-based tools. Nucleic Acids Research, 41, 590-596. https://doi.org/10.1093/nar/gks1219.

(2) Chen, S.F., Zhou, Y.Q., Chen, Y.R. and Gu, J. (2018). fastp: an ultra-fast all-in-one FASTQ preprocessor. Bioinformatics, 34(17), 884-890. https://doi.org/10.1093/bioinformatics/bty560.

(3) Magoc, T. and Salzberg, S.L. (2011). FLASH: fast length adjustment of short reads to improve genome assemblies. Bioinformatics, 27(21), 2957-2963. https://doi.org/10.1093/bioinformatics/btr507.

(4) Edgar, R.C. (2013). UPARSE: highly accurate OTU sequences from microbial amplicon reads. Nature Methods, 10(10), 996-+. https://doi.org/10.1038/Nmeth.2604.

(5) Wang, Q., Garrity, G.M., Tiedje, J.M. and Cole, J.R. (2007). Naive Bayesian classifier for rapid assignment of rRNA sequences into the new bacterial taxonomy. Applied and Environmental Microbiology, 73(16), 5261-5267. https://doi.org/10.1128/Aem.00062-07.

Why Silva database instead of PR2 database?

**Response:** Thank you very much for your questions and suggestions, and we will be willing to try to use the PR2 database for subsequent studies. The Silva database is a more comprehensive database (including bacteria, archaea, and eukaryotes), which has been widely used to annotate in different particle size ranges of microbial eukaryotes (micro-, pico-, and nanoeukaryotes) (Liu et al., 2021; Wang et al., 2021; Zhang et al., 2021). We also affirm that the PR2 database is curated specifically for protists. However, based on other references, the silva138 database is sufficient for our analysis of microbial eukaryotes, thus facilitating our comparison with other similar studies.

(1) Liu, Q., Zhao, Q., Jiang, Y., Li, Y., Zhang, C., Li, X., Yu, X., Huang, L., Wang, M., Yang, G., Chen, H. and Tian, J. (2021). Diversity and co-occurrence networks of picoeukaryotes as a tool for indicating underlying environmental heterogeneity in the Western Pacific Ocean. Mar Environ Res, 170, 105376. https://doi.org/10.1016/j.marenvres.2021.105376.
(2) Wang, F., Huang, B., Xie, Y., Cai, S., Wang, X. and Mu, J. (2021). Diversity, Composition, and Activities of Nano- and Pico-Eukaryotes in the Northern South China Sea With Influences of Kuroshio Intrusion. Frontiers in Marine Science, 8. https://doi.org/10.3389/fmars.2021.658233.
(3) Zhang, L., Yin, W., Wang, C., Zhang, A., Zhang, H., Zhang, T. and Ju, F. (2021). Untangling Microbiota Diversity and Assembly Patterns in the World's Largest Water Diversion Canal. Water Res, 204, 117617. https://doi.org/10.1016/j.watres.2021.117617.

Why OTUs instead of ASVs? Why QIIME1.9.1 instead of QIIME2?

**Response:** Thank you very much for your questions. As mentioned by the editor, both analyses (OTUs and ASVs) have their own characteristics, which have been discussed in previous studies (Amos et al., 2020; Glassman and Martiny 2018). Furthermore, we do not deny either analysis method.

The analysis to OTUs obtained using the UPARSE clustering method have still been widely used for high-throughput sequencing analysis (Gad et al., 2022; Reboul et al., 2021; Sun et al., 2022; Xu et al., 2022; Zhang et al., 2022). Some of the diversity indices in our study are more comparable to previous similar researches using 97% sequence similarity OTUs (Chen et al., 2022; Wang et al., 2021; Wang et al., 2020b), as comparisons of these statistics using the same bioinformatics tool still seem to remain persuasive, but the broad-scale ecological patterns remained robust

regardless of the feature-clustering method (Glassman and Martiny 2018; Li et al., 2019).

In addition, after preliminary analysis of the data based on ASVs, we found that compared to OTUs, the total number of sequences was reduced by 45% after ASVs picking and the number of normalized and rarified sequences was reduced by 35% (16717 vs. 10894).

We believe that the analysis of OTUs is appropriate for our current study and can also clearly describe our results. We have noted studies based on ASVs and do not exclude subsequent studies will use this approach. Most importantly, we never deny either analysis method.

We have revised in our manuscript. "*The analysis to OTUs obtained using the UPARSE clustering method have still been widely used for high-throughput sequencing analysis (Gad et al., 2022; Reboul et al., 2021; Sun et al., 2022; Xu et al., 2022; Zhang et al., 2022). Some of the diversity indices in our study were more comparable to previous similar researches using 97% sequence similarity OTUs (Chen et al., 2022; Wang et al., 2021; Wang et al., 2020b), as comparisons of these statistics using the same bioinformatics tool still seem to remain persuasive (Glassman and Martiny 2018; Li et al., 2019).*" (L196-202)

(1) Amos, G.C.A., Logan, A., Anwar, S., Fritzsche, M., Mate, R., Bleazard, T. and Rijpkema, S. (2020) Developing standards for the microbiome field. Microbiome 8(1). https://doi.org/10.1186/s40168-020-00856-3.

(2) Glassman, S.I. and Martiny, J.B.H. (2018) Broadscale Ecological Patterns Are Robust to Use of Exact Sequence Variants versus Operational Taxonomic Units. Ecological and Evolutionary Science 3(4), e00148-00118. https://doi.org/10.1128/mSphere.

(3) Gad, M., Hou, L., Cao, M., Adyari, B., Zhang, L., Qin, D., Yu, C.P., Sun, Q. and Hu, A. (2022) Tracking microeukaryotic footprint in a peri-urban watershed, China through machine-learning approaches. Science of the Total Environment 806(Pt 1), 150401. https://doi.org/10.1016/j.scitotenv.2021.150401.

(4) Reboul, G., Moreira, D., Annenkova, N.V., Bertolino, P., Vershinin, K.E. and Lopez-Garcia, P. (2021) Marine signature taxa and core microbial community stability along latitudinal and vertical gradients in sediments of the deepest freshwater lake. Isme Journal 15(11), 3412-3417. https://doi.org/10.1038/s41396-021-01011-y.

(5) Sun, P., Wang, Y., Huang, X., Huang, B.Q. and Wang, L. (2022) Water masses and their associated temperature and cross-domain biotic factors co-shape upwelling microbial communities. Water research 215. https://doi.org/10.1016/j.watres.2022.118274.

(6) Xu, D., Kong, H., Yang, E.J., Wang, Y., Li, X., Sun, P., Jiao, N., Lee, Y., Jung, J. and Cho, K.H. (2022) Spatial dynamics of active microeukaryotes along a latitudinal gradient: Diversity, assembly process, and co-occurrence relationships. Environ Res 212(Pt A), 113234.

https://doi.org/10.1016/j.envres.2022.113234.

(7) Zhang, W., Wan, W., Lin, H., Pan, X., Lin, L. and Yang, Y. (2022) Nitrogen rather than phosphorus driving the biogeographic patterns of abundant bacterial taxa in a eutrophic plateau lake. Science of the Total Environment 806(Pt 4), 150947. https://doi.org/10.1016/j.scitotenv.2021.150947.

(8) Chen, Z., Gu, T., Wang, X., Wu, X. and Sun, J. (2022) Oxygen gradients shape the unique structure of picoeukaryotic communities in the Bay of Bengal. Science of the Total Environment 814, 152862. https://doi.org/10.1016/j.scitotenv.2021.152862.

(9) Wang, F., Huang, B., Xie, Y., Cai, S., Wang, X. and Mu, J. (2021) Diversity, Composition, and Activities of Nano- and Pico-Eukaryotes in the Northern South China Sea With Influences of Kuroshio Intrusion. Frontiers in Marine Science 8. https://doi.org/10.3389/fmars.2021.658233.

(10) Wang, Y., Li, G., Shi, F., Dong, J., Gentekaki, E., Zou, S., Zhu, P., Zhang, X. and Gong, J. (2020) Taxonomic Diversity of Pico-/Nanoeukaryotes Is Related to Dissolved Oxygen and Productivity, but Functional Composition Is Shaped by Limiting Nutrients in Eutrophic Coastal Oceans. Front Microbiol 11, 601037. https://doi.org/10.3389/fmicb.2020.601037.

(11) Li, Y., Gao, Y., Zhang, W., Wang, C., Wang, P., Niu, L. and Wu, H. (2019) Homogeneous selection dominates the microbial community assembly in the sediment of the Three Gorges Reservoir. Science of the Total Environment 690, 50-60. https://doi.org/10.1016/j.scitotenv.2019.07.014.

(6) Line 161- here and elsewhere the OTUs index is referred to and I am not sure what this means. Perhaps you are referring to richness?

**Response**:We agreed with you that the information provided in "OTUs" is limited, even if operational taxonomic units (OTUs) do provide further details. The OTUs was used to describe richness in our manuscript. We have revised the "OTUs" to "*richness*" to reduce the confusion. (L222)

(7) 167- define MNTD at first use

**Response**:Thank you. We have already defined the MNTD at first use. "*mean nearest taxon distance (MNTD)*" (L229)

(8) 179- Bray-Curtis distance or dissimilarity, Not similarity

**Response**:The Bray- Curtis distance varies from 0 to 1, and we used distance to calculate similarity "similarity indices=1- distance indices". In order to reduce the confusion, we have already revised "All calculations were based on similarity matrices calculated with the Bray-Curtis similarity index." into "*All calculations were based on similarity matrices (1-dissimilarity of the Bray–Curtis distance metric).*" (L239)

(9) 181- I think you may want to scale these variables, especially for variance partitioning (z-scores)

**Response**:Yes, you are right. By transforming the data log(x+1), the effect of the magnitude can be reduced. Apart from the z-scores mentioned by the reviewer, the method used in this manuscript, i.e. log(x+1) transformation, was mostly used in similar studies.

(10) 219- a range of 0.9 to 7.14ºC does not feel similar

**Response**:Thank you. We have already revised in our manuscript. "*The water temperature (WT) of all five lakes varied from 0.90°C to 7.14°C, with…*" (L283-285)

(11) 222- molarity is mols per liter, so the units "uM L-1" is incorrect. Only uM should be reported.

**Response**:Revised.

(12) 226- the a of chlorophyll a should be italicized

**Response**:Revised.

(13) 227- salinity needs units (PSU?)

**Response**:Thank you. We have already revised in our manuscript and supplementary information.

(14) 223- the Good's coverage is calculated based on singletons, so please clarify that it was calculated before quality filtering. Also, providing rarefaction curves in the supplemental material will increase confidence in adequate sequencing depth and coverage.

**Response:** Before quality filtering, the Good's coverage can be calculated. Early in our analysis, the OTUs, classified as metazoa, and unassigned sequences, were filtered based on taxonomic metadata. And the sequences were normalized at the lowest sequence depth. The new Good's coverage will be calculated to judge if the libraries could represent most species in these samples. In our manuscript, the Good's coverage was obtained after quality filtering and all coverages were above 99% as required.
We have also already provided rarefaction curves in the supplemental material.

[Figure]

Fig. S1 Rarefaction curves of similarity-based operational taxonomic units (OTUs) at 97% sequence similarity level (a) and Shannon index(b).

(15) 246- SAR should be defined at its first appearance, I'm assuming stramenopiles-rhizaria-alveolates supergroup?

**Response:** Yes, you are right. We have revised the detailed information at its first appearance. "*Stramenopiles-Alveolates-Rhizaria (SAR)*" (L313)

(16) 251- 70.09% Arthropoda is …. a lot. Potentially fecal material since the samples were filtered through such fine mesh? I would consider excluding this sample unless the remaining sequencing reads still reach OTU saturation after the Arthropoda reads are removed. In general, you might consider removing metazoan reads early in the analysis.

**Response:** Agreed with you. It has been shown that, despite its shortcomings, filtration is the most feasible method for studying the diversity of eukaryotes size characteristics, and the method does reveal differences in the relative abundance of OTUs in different particle size ranges (Wang et al., 2021).

Based on the reviewer's suggestion, we re-analyzed the data by removing the metazoan sequences reads in advance of the analysis, and sequences were normalized at the lowest sequences depth and rarefied at 16,717 reads, yielding a total of 520 OTUs. This analysis did not affect the main results and conclusion of our study (including lower diversity, dominant taxa, co-occurrence network and assembly processes). The results, figures and discussion have been revised and more references were added in our manuscript.

(1) Wang, F., Huang, B., Xie, Y., Cai, S., Wang, X. and Mu, J. (2021). Diversity, Composition, and

Activities of Nano- and Pico-Eukaryotes in the Northern South China Sea With Influences of Kuroshio Intrusion. Frontiers in Marine Science, 8. https://doi.org/10.3389/fmars.2021.658233.

(17) 267- still unclear what the OTU index is

**Response:** The OTUs represent the richness. We have revised the "OTUs" to "*richness*" to clear the confusion.

(18) 285- define UPGMA at first use

**Response:** The detailed information on UPGMA was provided in L235. "*unweighted pair-group method with arithmetic means (UPGMA)*"

(19) 287- rather than saying "clustered into one clade," I think it is more correct to simply say "clustered together"

**Response:** Revised the "clustered into one clade" to "*clustered together*".

(20) 345- while it is true that the taxa found in the samples are small cells and their small size makes them better adapted to low nutrient conditions, I think that it is hard to say whether they were more or less abundant than larger cells since all the larger cells were removed by pre-filtering with 20 um mesh. As such, this section probably should not take up so much space or prominence in the discussion.

**Response:** Agreed with you. Chrysophyta was the predominant taxa among the microbial eukaryotes in the particle size range of our interest (0.2~20 μm). We have reduced the description of the small cells and revised the manuscript. "Firstly, the dominance may be due to the adaptation to low nutrient availability. *Chrysophyta has been well represented mostly in oligo and mesotrophic lakes from both the Maritime and Continental regions (Allende 2009; Allende and Izaguirre 2003; Izaguirre et al., 2020; Richards et al., 2005).*" (L423-425)

(1) Allende, L. (2009). Combined effects of nutrients and grazers on bacterioplankton and phytoplankton abundance in an Antarctic lake with even food-chain links. Polar Biology, 32(3), 493-501. https://doi.org/10.1007/s00300-008-0545-6.

(2) Allende, L. and Izaguirre, I. (2003). The role of physical stability on the establishment of steady states in the phytoplankton community of two Maritime Antarctic lakes. Hydrobiologia, 502(1-3), 211-224. https://doi.org/DOI 10.1023/B:HYDR.0000004283.11230.4a.

(3) Izaguirre, I., Allende, L. and Romina Schiaffino, M. (2020). Phytoplankton in Antarctic lakes: biodiversity and main ecological features. Hydrobiologia. https://doi.org/10.1007/s10750-020-04306-x.

(4) Izaguirre, I., Allende, L. and Romina Schiaffino, M. (2020). Phytoplankton in Antarctic lakes: biodiversity and main ecological features. Hydrobiologia.

https://doi.org/10.1007/s10750-020-04306-x.

(5) Richards, T.A., Vepritskiy, A.A., Gouliamova, D.E. and Nierzwicki-Bauer, S.A. (2005). The molecular diversity of freshwater picoeukaryotes from an oligotrophic lake reveals diverse, distinctive and globally dispersed lineages. Environ Microbiol, 7(9), 1413-1425. https://doi.org/10.1111/j.1462-2920.2005.00828.x.

(21) 363- I am not sure what is meant by "forming temporary groups"— maybe change the word choice?

**Response:** We have supplied detailed information. Revised as "by forming temporary groups" to "*by forming temporary, non-swimming cell populations encased in a gelatinous mother cell membrane.*" (L443-444)

(22) 378- please clarify whether the other studies you are referring to used similar size fractionation

**Response:** Thanks for your question. We have confirmed the lower diversity in our study compared with other similar studies mentioned above, which use similar size fractionation. Also, more references were supplied. "*Compared with other aquatic ecosystems* (Hernandez-Ruiz et al., 2018; Wang et al., 2021; Wang et al., 2020b)*, the diversity…*". (L458-460)

(1) Wang, F., Huang, B., Xie, Y., Cai, S., Wang, X. and Mu, J. (2021). Diversity, Composition, and Activities of Nano- and Pico-Eukaryotes in the Northern South China Sea With Influences of Kuroshio Intrusion. Frontiers in Marine Science, 8. https://doi.org/10.3389/fmars.2021.658233.

(2) Wang, Y., Li, G., Shi, F., Dong, J., Gentekaki, E., Zou, S., Zhu, P., Zhang, X. and Gong, J. (2020). Taxonomic Diversity of Pico-/Nanoeukaryotes Is Related to Dissolved Oxygen and Productivity, but Functional Composition Is Shaped by Limiting Nutrients in Eutrophic Coastal Oceans. Front Microbiol, 11, 601037. https://doi.org/10.3389/fmicb.2020.601037.

(3) Hernandez-Ruiz, M., Barber-Lluch, E., Prieto, A., Alvarez-Salgado, X.A., Logares, R. and Teira, E. (2018). Seasonal succession of small planktonic eukaryotes inhabiting surface waters of a coastal upwelling system. Environ Microbiol, 20(8), 2955-2973. https://doi.org/10.1111/1462-2920.14313.

(23) 426- "indicating that species coexistence was achieved mainly by symbiotic relationships between species" — I think this is an overstatement and not supported by the data.

**Response:** We have revised in the discussion. "By analyzing the network, we found that the positive correlations were much more than the negative correlations in the co-occurrence network (82% vs. 18%), *revealing that positive interaction (e.g., due to cross-feeding, niche overlap, mutualism, and/or commensalism) might exhibit a more*

*important role than negative interaction (e.g., predator-prey relationships, host-parasite relationships and/or competition)* (Chen and Wen 2021) *in studied Antarctic lake ecosystem. A similar result has been found in small planktonic eukaryotes (0.2~20 μm) inhabiting the surface waters of a coastal upwelling system (Hernandez-Ruiz et al., 2018). Notwithstanding, further studies are necessary to corroborate the biological interactions and other nonrandom processes (for example, cross-feeding versus niche overlap) between species pairs detected by network analyses.* (L515-524)

(1) Chen, W. and Wen, D. (2021). Archaeal and bacterial communities assembly and co-occurrence networks in subtropical mangrove sediments under Spartina alterniflora invasion. Environ Microbiome, 16(1), 10. https://doi.org/10.1186/s40793-021-00377-y.

(2) Hernandez-Ruiz, M., Barber-Lluch, E., Prieto, A., Alvarez-Salgado, X.A., Logares, R. and Teira, E. (2018). Seasonal succession of small planktonic eukaryotes inhabiting surface waters of a coastal upwelling system. Environ Microbiol, 20(8), 2955-2973. https://doi.org/10.1111/1462-2920.14313.

(24) 465- unclear which "channel" is being referred to, more context is needed

**Response:** Revised the "channel" to "*Middle Route Project of the South-to-North Water Diversion Projects in China.*" (L565-566)

(25) 478- what is "ecological scheduling"?

**Response:** We have deleted this description.

(26) 484- the statement regarding extreme conditions exerting less selection pressure seems incorrect?

**Response:** Thank you and we agreed with you! We have deleted this incorrect description.

(27) 511- again please be careful about assuming that positive co-occurrence patterns equate to symbioses, it seems niche-overlapping is more likely

**Response:** Yes. We agreed that this assuming is not suitable. Further studies are necessary to corroborate the biological interactions and other nonrandom processes (for example, cross-feeding versus niche overlap) between species pairs detected by network analyses.

(28) 520- please provide the PRJ number to make it easier to access all the sequences.
**Response:** Yes, we agreed and provided the PRJ number. "*PRJNA805287*" (L628)

(29) Figure 1- Please also include a large map that places the region in regional context (probably include Antarctic peninsula and tip of South America)

**Response:** Yes, we have provided a large map. (Fig.1)

[Figure]

(30) Figure 3-The significance indications of letters are not defined in the figure caption. What do "a", "b", and "ab" mean?

**Response:** We agreed with you that the information provided in " "a", "b", and "ab" mean" is limited. "*The significant differences (P<0.05) were indicated by different alphabet letters between lakes, and lakes contained the same alphabet letters showed no significant difference (P>0.05).*" (L1113-1115)

**Other comments**

English editing is needed throughout the manuscript. Below are a few edits that stood out to me.

(1) 125- nutrient to nutrients

**Response:** Thanks, we have revised these errors from the figure and the rest of the

text.

(2) 189- opening sentence needs to be rewritten

**Response:** The opening sentence has been rewritten as "*We constructed one co-occurrence network based on samples from the whole study period.*" (L252-253)

(3) 190- "OTUs represented Occurred"?

**Response:** Revised as "*OTUs occurred*".

(4) 217- Result to Results

**Response:** Revised.

(5) 350- "still keeps a high proportion" needs to be reworded

**Response:** Revised as "*still retains high cell density*". (L428)

(6) 353 reference mistake "F R Pick"- remove 1st initials

**Response:** We are very sorry for the error, and we have revised citation of this reference "*Pick and Lean 1984*". (L433)

(7) 406- "the nonconsecutive of environmental factors among different expedition seasons was deficient in our study" as is, I cannot make out the meaning of this sentence.

**Response:** It was very difficult to obtain all the environmental factors during our expedition. The unexplained community variation in this study could also be due to the absence of environmental factors that were not fully obtained (Wang et al., 2020a). Future research should strive to obtain and consider more environmental factors. For a better understanding, we have revised this sentence to "*Firstly, it is not easy to obtain all environmental factors, and some important factors may exist that are not fully obtained or taken into account in the current study.*" (L493-495)

(1) Wang, W., Ren, K., Chen, H., Gao, X., Ronn, R. and Yang, J. (2020) Seven-year dynamics of testate amoeba communities driven more by stochastic than deterministic processes in two subtropical reservoirs. Water Res 185, 116232. https://doi.org/10.1016/j.watres.2020.116232.

---

## Referee Report (RR1)

The authors did a nice job adequately addressing most of the reviewers' comments. In particular, the methods are much more clear now. However, the writing still needs some work. There are some sentence fragments and run-on sentences, some citations seem incorrect or out of place, and in many places it is difficult to understand the authors' argument. I have made some suggestions below, but I recommend additional editing and revising.

Line 1 (Title) - Suggestion - Diversity and assembly processes of microbial eukaryote communities in Fildes Peninsula Lakes (West Antarctica)

Line 24: ranges should be indicated by an en dash, not a tilde.

Line 25: when "lake" is used as part of a proper name, it is capitalized, e.g., "lake Chang Hu" should be "Lake Chang Hu".

Line 25: …with higher values in Lake Chang Hu and Lake Kitec and the lowest value in Lake Yue Ya.

Line 37: The stochastic processes... to Stochastic processes ("the" is unnecessary - check articles throughout the manuscript, "the" is often used when "a" or no article is more appropriate).

Stochastic processes (e.g., homogenizing dispersal and undominated process) dominated community assembly compared to deterministic processes.

Line 39: These findings demonstrate the diversity of microbial eukaryotic communities in the freshwater lakes of the Fildes Peninsula and have important implications for understanding community assembly in these ecosystems.

Line 45: A suggestion: The Fildes Peninsula--which makes up the southwestern end of King George Island, South Shetland Islands, Antarctica--is home to a relatively high density of scientific research stations. The peninsula is commonly ice-free throughout the austral summer, making it the largest ice-free area (40 $km^2$) on King George Island. Falling within the maritime Antarctic, the peninsula experiences 400-600 mm of precipitation each year and has an average annual(??) temperature of -3ºC. Nevertheless, permafrost and periglacial processes can be found in the region. Lakes on the Fildes Peninsula, along with those found in other ice-free areas of Antarctica, represent the year-round liquid water reservoirs on the continent. Water availability and quality are impacted by sea conditions, macro-fauna usage, and anthropogenic influences, such as solid, volatile, and fluid waste production and disposal. Antarctic lake systems are sentinels for climate change and host globally-relevant microbes and biogeochemical cycles, thus making a more complete understanding of the processes shaping microbial communities there a priority. Moreover, the physical stability observed in these lakes makes them a good model system for interrogating biogeochemical processes within water columns.

Line 61: ranges should be indicated by an en dash, not a tilde.

Line 68: A suggestion: However, microbial eukaryotes in Antarctic lakes have been understudied due to their small cell size and lack of conspicuous morphological features.

Microbial Eukaryotes have not been neglected in all systems - this sentence needs more specificity to be true. Even in the revised form, I suspect many people would object to the statement. Maybe it is better to say that they are less studied than bacteria or zooplankton.

A better (?) suggestion:

Microbial eukaryotes are generally difficult to study due to their small size and common lack of distinguishing morphological features, especially among the pico- and nanoeukaryotes.

Line 79: reflection of environmental conditions

.
.
.

(needs significant editing for clarity and readability)

.
.
.

Methods

How was water collected for Chl *a* and nutrient analysis? Niskin bottle? What volume was filtered through what kind of filter (GFF?) for acetone extracted Chl *a*?

Line 223: the "respectively" is unnecessary

Line 225: OTUs occurring in at least five samples

Line 258: Bastian M et al., 2009 should just be Bastian et al., 2009 ?

Line 261: neutral should not be capitalized

Results

Lines 283-290 should be rewritten for clarity. There are fragments and run-ons. Again, tildes are not appropriate for ranges.

Line 294: Diveristy and composition of microbial eukaryotic communities

Line 329: over the years  [because there is only one sample per year)
Line 332: years
Line 361-2: clustered separately.

Line 371: variablity among lakes, but there was still a large amount of unexplained variation.
Line 373: made up the microbial eukaryotic community network

Discussion
Line 416: differs

Line 425: the maritime and continental regions of Antarctica? If not, then remove "the". If yes, say it.
Line 430-437: I think the present tense was correct here

Line 465: states that there should be an increased species number as habitat area increases with a specific area

Line 471: Previous studies have demonstrated
Line 472-473: In this study, we found

Line 477: I would keep water temperature as water temperature throughout the discussion

Line 492: A substantial amount of variation was unexplained, which could be due to a number of reasons. First, it is not possible to measur all environmental factors that can influence microbial communities and, thus, some significant driving factors may not have been included in the study. Potentially vital abiotic factors in Antarctic lakes include: solar cycle, light availability, ice cover (thickness and duration), physical mixing of snow melt, and other hydrological processes. Second, relationships between microorganisms could not be quantified, and these relationships are potentially essential factors shaping community structure. For example, predation pressure can manifest as a top-down control of microbial eukaryotes. Third, …

Line 507: Network analysis can help illuminate complex biological interactions
Line 514: we found that positive correlations were much more common (82%) than negative correlations (18%). These results suggest that positive relationships (e.g., due to cross-feeding, niche overlap, mutualism, or commensalism) might play a more important role in Antarctic lake ecosystems than negative relationships (e.g., predator-prey, host-parasite, or competition).

527: the clause "and these might weaken the role of environmental selection in community assembly" does not make sense

528: The sentence starting with "Previous studies have shown the high response of microbial … " seems completely unrelated to the rest of the paragraph

545 and 549 - I think this should be present tense

552: from aquatic ecosystems

552: For example, picoeukaryotic communities in the lower …

554: Results from our study supported a more prominent role for stochastic processes than deterministic in shaping the assembly of microbial eukaryotic communities.

557: a small amount of variation in our

565: It is still unclear what is being referred to and why it is significant - I googled what the Middle Route Project of the South-toNorth Water Diversion Project is and it does not seem to be related to lakes in polar ecosystems, why did you choose this citation? There must be something published that is more relevant and can better put your findings in context?

Undominant processes should be better defined?I think a better description would help readers.

617: "were proved" seems too strongly worded. Maybe: Stochastic processes and biotic co-occurrence patterns were shown to be important in shaping microbial eukaryotic communiitees in the area.

621: the sentence starting with "Stochastic processes played a very prominent … " is repetitive and should be removed or edited

---

## Author Response (AR2)

Dear editor,

We are deeply grateful for the efforts of you and reviewers to improve the quality of our manuscript. We have made our efforts to revise the manuscript with clarifications/elaborations as following.

In this version, we have made revisions based on the comments: (1) adjusted some content, incorporate the suggested edits, and carefully edit the entire manuscript for grammar and clarity. (2) adjusting reference list based on manuscript preparation guidelines. Meanwhile, the manuscript has been polished by proof reading service (**https://www.proof-reading-service.com/en/**). A list of all the changes made can be found in the point-by-point response to the reviewers' comments.

**Our response** is in **normal font** and colored in **blue**, and *the revised text* is in *italic font* and colored in ***blue***. Lines number refer to the track changed version.

**General Response to Reviewer 2 Comments**

The authors did a nice job adequately addressing most of the reviewers' comments. In particular, the methods are much more clear now. However, the writing still needs some work. There are some sentence fragments and run-on sentences, some citations seem incorrect or out of place, and in many places it is difficult to understand the authors' argument. I have made some suggestions below, but I recommend additional editing and revising.

Thanks for your comments and suggestions. We are appreciated with your helpful advice, and we have made our efforts to revise the manuscript with clarifications/elaborations as following.

**Our response** is in **normal font** and colored in **blue**, and *the revised text* is in *italic font* and colored in ***blue***. Lines number refer to the track changed version.

(1) Line 1 (Title) - Suggestion - Diversity and assembly processes of microbial eukaryote communities in Fildes Peninsula Lakes (West Antarctica)

**Response:** Revised. "community" to "*communities*"

(2) Line 24: ranges should be indicated by an en dash, not a tilde.

**Response:** Revised. In the manuscript, we revised all descriptions indicating ranges, and "~" was replaced with "-".

(3) Line 25: when "lake" is used as part of a proper name, it is capitalized, e.g., "lake Chang Hu" should be "Lake Chang Hu".

**Response:** Revised in the manuscript.

(4) Line 25: …with higher values in Lake Chang Hu and Lake Kitec and the lowest value in LakeYue Ya.

**Response:** Revised. "*…with higher values recorded in Lake Chang Hu and Lake Kitec and the lowest value obtained for Lake Yue Ya*" Line 25-26

(5) Line 37: The stochastic processes... to Stochastic processes ("the" is unnecessary – check articles throughout the manuscript, "the" is often used when "a" or no article is more appropriate).
Stochastic processes (e.g., homogenizing dispersal and undominated process) dominated community assembly compared to deterministic processes.

**Response:** Agreed and revised. "*Stochastic processes (e.g., homogenising dispersal and undominated processes) predominated in community assembly over the deterministic processes*" Line 34-36

(6) Line 39: These findings demonstrate the diversity of microbial eukaryotic communities in the freshwater lakes of the Fildes Peninsula and have important implications for understanding community assembly in these ecosystems.

**Response:** Agreed and revised according to your advice. Line 35-38

(7) Line 45: A suggestion: The Fildes Peninsula--which makes up the southwestern end of King George Island, South Shetland Islands, Antarctica--is home to a relatively high density of scientific research stations. The peninsula is commonly ice-free throughout the austral summer, making it the largest ice-free area (40 km2) on King George Island. Falling within the maritime Antarctic, the peninsula experiences 400-600 mm of precipitation each year and has an average annual(??) temperature of -3ºC. Nevertheless, permafrost and periglacial processes can be found in the region. Lakes on the Fildes Peninsula, along with those found in other ice-free areas of Antarctica, represent the year-round liquid water reservoirs on the continent. Water availability and quality are impacted by sea conditions, macro-fauna usage, and anthropogenic influences, such as solid, volatile, and fluid waste production and disposal. Antarctic lake systems are

sentinels for climate change and host globally-relevant microbes and biogeochemical cycles, thus making a more complete understanding of the processes shaping microbial communities there a priority. Moreover, the physical stability observed in these lakes makes them a good model system for interrogating biogeochemical processes within water columns.

**Response:** Agreed and revised according to your advice. Line 44-70

(8) Line 61: ranges should be indicated by an en dash, not a tilde.

**Response:** Revised as suggested.

(9) Line 68: A suggestion: However, microbial eukaryotes in Antarctic lakes have been understudied due to their small cell size and lack of conspicuous morphological features. Microbial Eukaryotes have not been neglected in all systems - this sentence needs more specificity to be true. Even in the revised form, I suspect many people would object to the statement. Maybe it is better to say that they are less studied than bacteria or zooplankton.
A better (?) suggestion:
Microbial eukaryotes are generally difficult to study due to their small size and common lack of distinguishing morphological features, especially among the pico- and nanoeukaryotes.

**Response:** Thanks for a better suggestion. We have revised the manuscript according to your advice. Line 79-81

(10) Line 79: reflection of environmental conditions

**Response:** Revised.

**Methods**

(1) How was water collected for Chl a and nutrient analysis? Niskin bottle? What volume was filtered through what kind of filter (GFF?) for acetone extracted Chl a?

**Response:** We have made a detailed addition in the Method. "*In addition to the in-situ measurements, a water sample obtained 0.5 m below the surface was collected at each monitoring site using a 5 L plexiglass sampler. For measurements of chlorophyll a (Chl a) contents, 1 L water samples were filtered using GF/F filters (0.70 μm,*

*Whatman), and Chl a was then extracted with 90% acetone over 24 h and measured spectrophotometrically*" Line 170-174

(2) Line 223: the "respectively" is unnecessary

**Response:** Deleted.

(3) Line 225: OTUs occurring in at least five samples

**Response:** Revised. "occurred" to "*occurring*" Line 264

(4) Line 258: Bastian M et al., 2009 should just be Bastian et al., 2009 ?

**Response:** Revised. Line 267

(5) Line 261: neutral should not be capitalized

**Response:** Revised.

**Results**

(1) Lines 283-290 should be rewritten for clarity. There are fragments and run-ons. Again, tildes are not appropriate for ranges.

**Response:** Revised. "*Water temperature ranged from 0.90°C to 7.14°C, and the water temperature in YO was significantly higher than in the other lakes (Table S1, P<0.05). Nutrients in five lakes showed lower values of 0.00-0.15 μM (NO$_2^-$), 0.05-0.74 μM (NH$_4^+$) and 0.02-2.29 μM (PO$_4^{3-}$). Relatively higher and lower levels of nutrients were identified in YY and XH, respectively. The lowest value of SiO$_3^{2-}$ was recorded in YY (1.43 μM) and the highest in CH (51.5 μM). The highest and lowest Chl a were reached in YY and CH (2.11 and 0.25 μg L$^{-1}$, respectively). pH showed minimum and maximum values in YY (7.65) and CH (8.27). Sal values ranged from 0.00-0.14 PSU and were significantly lower in YO than in the other lakes (P<0.05).*" Line 292-300

(2) Line 294: Diveristy and composition of microbial eukaryotic communities

**Response:** Revised. "*Diversity and composition of microbial eukaryotic communities*" Line 310

(3) Line 329: over the years [because there is only one sample per year)

**Response:** Revised. Line 335

(4) Line 332: years

**Response:** Revised. Line 337

(5) Line 361-2: clustered separately.

**Response:** Revised. Line 365

(6) Line 371: variablity among lakes, but there was still a large amount of unexplained variation.

**Response:** Revised. Line 375-376

(7) Line 373: made up the microbial eukaryotic community network

**Response:** Revised. Line 377

**Discussion**

(1) Line 416: differs

**Response:** Revised. "*The environmental conditions (e.g., low light and low nutrient contents, etc.) in Antarctic freshwater lakes differ from those of temperate lakes*" Line 415-416

(2) Line 425: the maritime and continental regions of Antarctica? If not, then remove "the". If yes, say it.

**Response:** Thanks for your suggestions. Revised as "*the maritime and continental regions of Antarctica*" Line 427

(3) Line 430-437: I think the present tense was correct here

**Response:** We agreed with your advice. Revised.

(4) Line 465: states that there should be an increased species number as habitat area increases with a specific area

**Response:** Revised according to your advice. Line 468-469

(5) Line 471: Previous studies have demonstrated

**Response:** Revised. Line 476

(6) Line 472-473: In this study, we found

**Response:** Revised. Line 478

(7) Line 477: I would keep water temperature as water temperature throughout the discussion

**Response:** Thanks for your suggestion. We agreed with you. We keep water temperature as water temperature in discussion and results. But we still use the WT as water temperature in Figures.

(8) Line 492: A substantial amount of variation was unexplained, which could be due to a number of reasons. First, it is not possible to measure all environmental factors that can influence microbial communities and, thus, some significant driving factors may not have been included in the study. Potentially vital abiotic factors in Antarctic lakes include: solar cycle, light availability, ice cover (thickness and duration), physical mixing of snow melt, and other hydrological processes. Second, relationships between microorganisms could not be quantified, and these relationships are potentially essential factors shaping community structure. For example, predation pressure can manifest as a top-down control of microbial eukaryotes. Third, …

**Response:** Thanks for your suggestion. We agreed with you and revised the manuscript as suggested. Line 498-514

(9) Line 507: Network analysis can help illuminate complex biological interactions

**Response:** Revised. "could" to "*can*". Line 521

(10) Line 514: we found that positive correlations were much more common (82%) than negative correlations (18%). These results suggest that positive relationships (e.g., due to cross-feeding, niche overlap, mutualism, or commensalism) might play a more important role in Antarctic lake ecosystems than negative relationships (e.g., predator-prey, host-parasite, or competition).

**Response:** Revised as suggested. "*Our network analysis revealed that the positive correlations were much more common (82%) than negative ones (18%). These results*

*suggested that positive relationships (e.g., due to cross-feeding, niche overlap, mutualism, or commensalism) might play a more important role in the lake ecosystems of Antarctica than negative relationships (e.g., predator-prey, host-parasite, or competition)*" Line 528-534

(11) 527: the clause "and these might weaken the role of environmental selection in community assembly" does not make sense

**Response:** Thanks for your comments. We deleted this clause.

(12) 528: The sentence starting with "Previous studies have shown the high response of microbial … " seems completely unrelated to the rest of the paragraph

**Response:** Thanks for your comments. We deleted this sentence.

(13) 545 and 549 - I think this should be present tense

**Response:** Revised. "existed" to "*exist*", "influenced" to "*can influence*". Line 550 and 554

(14) 552: from aquatic ecosystems

**Response:** Revised. "from the aquatic ecosystems" to "for other microbial eukaryotic communities *in aquatic ecosystems*". Line 557

(15) 552: For example, picoeukaryotic communities in the lower …

**Response:** Revised. "The picoeukaryotic communities in the lower …" to "*For example, picoeukaryotic communities in the lower …*". Line 558

(16) 554: Results from our study supported a more prominent role for stochastic processes than deterministic in shaping the assembly of microbial eukaryotic communities.

**Response:** Revised as suggested. Line 559-561

(17) 557: a small amount of variation in our

**Response:** Revised. "a small number of variations…" to "*a small amount of variation…*" Line 564-565

(18) 565: It is still unclear what is being referred to and why it is significant - I googled

what the Middle Route Project of the South-to North Water Diversion Project is and it does not seem to be related to lakes in polar ecosystems, why did you choose this citation? There must be something published that is more relevant and can better put your findings in context?

**Response:** Thanks for your suggestion. The South-North Water Transfer Project represents a special kind of freshwater ecosystem. Previous studies of eukaryotic microorganisms in this region using neutral community model (NCM) have similar results to our study.

Considering that the South-North Water Transfer Project is indeed not a polar ecosystem, we deleted the citation to this literature.

(19) Undominant processes should be better defined? I think a better description would help readers.

**Response:** Thanks for your suggestion. Undominant processes mostly consist of diversification, drift, weak selection, and weak dispersal, but it is still a challenge to directly divide into different components (Zhou and Ning, 2017; Stegen et al., 2015). For better understanding, we have indeed defined "Undominant processes" in detail in the material and method. "and undominated processes (i.e., weak selection, weak dispersal, diversification, and drift processes) with $|\beta NTI| < 2$ and $|RC_{bray}| < 0.95$." Line 286-287

(1) Stegen, J. C., Lin, X. J., Fredrickson, J. K., and Konopka, A. E.: Estimating and mapping ecological processes influencing microbial community assembly, Frontiers in Microbiology, 6, https://doi.org/10.3389/fmicb.2015.00370, 2015.

(2) Zhou, J. Z. and Ning, D. L.: Stochastic Community Assembly: Does It Matter in Microbial Ecology?, Microbiol Mol Biol R, 81, https://doi.org/10.1128/MMBR.00002-17, 2017.

(20) 617: "were proved" seems too strongly worded. Maybe: Stochastic processes and biotic co-occurrence patterns were shown to be important in shaping microbial eukaryotic communities in the area.

**Response:** Thanks for your comments. We agreed with you and revised according to your advice. Line 608-609

(21) 621: the sentence starting with "Stochastic processes played a very prominent … " is repetitive and should be removed or edited

**Response:** Thanks for your comments. We removed the sentence.

---

## Author Response (AR3)

Dear editor,

We are deeply grateful for the efforts of you and reviewers to improve the quality of our manuscript. We have made our efforts to revise the manuscript with clarifications/elaborations as following. A list of all the changes made can be found in the point-by-point response to your comments.

**Our response** is in **normal font** and colored in **blue**, and *the revised text* is in *italic font* and colored in **blue**.

**Comments to the author**:

This manuscript is much improved but still has some sentences that are not currently correct. These will need to be fixed before I can recommend for publication. Line numbers correspond to the trach changes version of the MS:

(1) L 293 – Nutrients in the sampled lakes were in general quite low in concentration with values of …

**Response:** Revised. "Nutrients in five lakes showed lower values of…" to "*Nutrients in the sampled lakes were in general quite low in concentration with values of…*"

(2) L299 Sal should be spelled out to Salinity

**Response:** Revised. "Sal" to "*Salinity*"

(3) L338 YO_18 "compared to other samples"

**Response:** Revised. "than in the other samples" to "*compared to other samples*"

(4) L340 significantly "lower relative abundance" than in YO…

**Response:** Revised. "significantly lower than in YO" to "*significantly lower relative abundance than in YO*".

(5) L343; species richness was highest in 2017 and lowest in 2018.

**Response:** Revised. "; the species richness were highest during the expedition season 2017 and lowest in 2018" to "; *species richness was highest in 2017 and lowest in 2018*"

(6) L349 "exhibited a relatively"

**Response:** Revised. *"exhibited relatively" to "exhibited a relatively"*

(7) L380 delete with

**Response:** Revised. *"comprising with 81.82%" to "comprising 81.82%"*

(8) L464 – "The diversity of the microorganisms reported here decreases from midlatitude to the poles." This study did not sample from mid latitude to the poles. This statement is incorrectly modified.

**Response:** Thanks for your comments. We agreed with you and revised it. The diversity of the five lakes studied was lower by comparing with other regions. The previous revision was incorrect; thus, we have removed this sentence.
(9) L469 – with => within

**Response:** "with a specific area" to "*within a specific area*"
(10) L485 warming has the risk of reducing the abundance and diversity of microorganisms,

**Response:** Revised. "warming had the…" to "*warming has the…*"
(11) L543 Above the role of environmental conditions is identified on community structure and here it is said to not be important. I'd suggest "at the OTU level only 8% of the taxa correlated with environmental conditions, identifying these factors act on the community level while maybe not the

**Response:** Thanks for your comments. We agree with you. This sentence was confusing and we deleted it.
(12) L594 – if the lakes were covered by ice that would reduce transmission but a limited geographical range would increase the transmission rate. The first and second half of this sentence point to different factors for dispersal so the sentence should be modified.

**Response:** Thanks for your comments. Dispersal is considered limited in cases where microbial migration in space is restricted and/or impeded (Zhou and Ning, 2017). The five studied lakes are disconnected, and as the long geographical distance between the lakes here, resulting in the limited dispersal of microorganisms.
To improve clarity, we have modified the sentence to "*In addition, the five studied lakes were covered in ice for most of the year and there were long geographical distances between the lakes, resulting in the limited dispersal of microorganisms (0.95%).*"

1) Zhou, J. Z. and Ning, D. L.: Stochastic Community Assembly: Does It Matter in Microbial Ecology?, Microbiol. Mol. Biol. Rev., 81, https://doi.org/10.1128/MMBR.00002-17, 2017.